# Response of *Pinus sylvestris* var. *mongolica* to water change and drought history reconstruction in the past 260 years, northeast China

Liangjun Zhu[1, 2], Qichao Yao[1], David J. Cooper[2], Shijie Han[3,4], Xiaochun Wang[1*]

[1]School of Forestry, Northeast Forestry University, Harbin 150040, China;

[2]Department of Forest and Rangeland Stewardship, Colorado State University, Fort Collins, CO 80523, USA;

[3]School of Life Science, Henan University, Kaifeng 475004, China

[4]Institute of Applied Ecology, Chinese Academy of Sciences, Shenyang 110016, China

*Correspondence to*: Xiaochun Wang (wangx@nefu.edu.cn)

**Abstract.** We present a 260-year annual PDSI reconstruction based on a tree-ring width chronology of Scots pine (*Pinus sylvestris* var. *mongolica*) from four sample sites in the Central Daxing'an Mountains, northeast China. The reconstruction equation explained 38.2 % of the variance of annual PDSI in the calibration period from 1911 to 2010. Our reconstruction confirmed the local historical documents and other nearby hydroclimate reconstructions. Drought in the 1920s-1930s was more severe in the Daxing'an Mountains than the surrounding areas. A slight moisture increase was identified in the study area, while a warm-dry pattern was found in the West-Central Mongolian Plateaus (mild drier) and their transition zones: The West-Central Mongolian Plateaus (severe drier). Overall, the variation of drought in the Daxing'an Mountains and its relationship with surrounding areas may be affected by the Pacific or Atlantic oscillations (e.g., ENSO, PDO, AMO, NAO and SNAO), which can affect the Asian monsoon, change the local temperature and precipitation, and lead to drought.

**Keywords**   PDSI reconstruction; Central Daxing'an Mountains; Tree rings; *Pinus sylvestris* var. *mongolica*; PDO; AMO; Drought

## 1 Introduction

Drought as one of the major natural disasters is being more frequently with climate change in the world (Cook et al., 2010; Dai, 2011, 2013; Davi et al., 2006; Li et al., 2016). Severe droughts can threaten agriculture and human social activities, and also have a devastating impact on human lives and the survival of native and domestic plants and animals (Cook et al., 2010; Dong et al., 2013; Shen, 2008; Sun, 2007). Drought is one of the most severe and frequent natural disasters in China, especially in semi-arid and arid regions (Bao et al., 2015; Chen et al., 2015; Cook et al., 2010; Dong et al., 2013; Liang et al., 2006; Shen, 2008; Sun and Liu, 2013; Xu, 1998). For example, the drought in the 1920s affected almost all of northern China, accompanied by severe economic and social losses (Dong et al., 2013; Liang et al., 2006; Shen, 2008; Sun, 2007). Natural droughts are recorded in tree rings in the arid or semi-arid regions (Bao et al., 2015; Chen et al., 2015; Liang et al., 2006; Sun and Liu, 2013; Wang and Song, 2011). Recent studies indicate a trend of increasing drought frequency, persistence and severity due to global warming in many regions of the world (Bao et al., 2015; Cook et al., 2010; Dai, 2011, 2013; Schrier et al., 2013). A rapid and pronounced warming accompanied by a decrease in precipitation has occurred in China, especially in high latitude and high altitude regions (Bao et al., 2015; Chen et al., 2015; Cook et al., 2010; Dai, 2013; Sun and Liu, 2013; Zhu et al., 2017), leading to severe and prolonged drought in recent decades, such as from 1999 to 2002 (Bao et al., 2015; Liu et al., 2009; Shen, 2008).

The Daxing'an Mountains in northeast China is a transition area from semi-humid climate in the east to more arid conditions in the west. (Bao et al., 2015; Zhao et al., 2002). The Asian monsoon

system directly affects the occurrence, intensity and severity of droughts and floods (Bao et al., 2015; Cook et al., 2010; Liang et al., 2006; Wang et al., 2013; Wang et al., 2005; Zhao et al., 2002) that has a devastating effects on human society and economy as well as natural ecosystems (Sun, 2007; Xu, 1998). For example, the drought in 2009 affected 81 million people in northeast China and more than 720,000 hectares of farmland suffered from water shortages (http://www.chinadaily.com.cn/cndy/2009-08/13/content_8562996.htm). In addition, drought also can increase the occurrence of large wildfires. Drought in Daxing'an Mountains, especially in spring or early summer, often leads to high risk of forest wildfires (Sun, 2007). The fire caused by drought in northern Daxing'an Mountains in May 1987 killed over 200 people and burned ~17,000 km$^2$ (Sun, 2007; Yao et al., 2017).

To better characterize current drought conditions and project those of the future, an improved understanding of past drought variability and potential forcing mechanisms is necessary. However, the shorter meteorological records in Daxing'an Mountains only started in 1950s limited our understanding of the long-term regime of past droughts. Tree rings can serve as an important high resolution proxy for long-term drought reconstructions (Cook et al., 2010; Dai, 2011; Pederson et al., 2013), and several hydroclimate reconstructions (Bao et al., 2015; Lv and Wang, 2014; Wang and Lv, 2012) have been conducted in northern China. Cook et al. (2010) also reconstructed the June-July-August Dai-PDSI in 534 grid points (Monsoon Asia Drought Atlas, MADA) in monsoon Asia using 327 tree-ring width chronologies. However, some disagreements occur between the MADA and the tree-ring-based local drought reconstructions and instrumental drought data, especially in eastern Asia, which might be an insufficient tree-ring data in eastern Asia used in MADA (Li et al., 2015; Liu et al., 2017). Additional drought reconstructions in eastern Asia are needed to gain a more thorough understanding of the

variability in the Asian Monsoon. Many researchers use the Palmer drought severity index (PDSI),
calculated from a water balance equation, incorporating air temperature and precipitation, to estimate
drought periodicity and intensity (Bao et al., 2015; Cook et al., 2010; Dai, 2011; Sun and Liu, 2013).
Here, we present a 260-year reconstruction of annual PDSI using tree-ring chronologies from the
Daxing'an Mountains to identify the timing of droughts and their correlation with eastern Mongolian
Plateaus climate as well as their potential forcing mechanisms.

**2 Materials and methods**
**2.1 Study area**
The Daxing'an Mountains, in northeast Inner Mongolia and northwest Heilongjiang Province, form
an important natural geographic divide between the Pacific Ocean and the north-western semi-arid
inland (Fig. 1). It is known to be a transition zone from the semi-humid to semi-arid region or from a
monsoon to non-monsoon climates (Zhao et al., 2002). The summer monsoons from the south-east are
blocked by the mountains and cannot penetrating further to the northwest. The western region is more
arid, and the eastern region is wetter. Summer weather is clarified by periodic incursions of warm,
humid air masses from low-latitude oceans, while dry and cold air in winter persists air masses invade
from high latitudes.
This study was conducted in high-latitude forests in the Daxing'an Mountains, northeast China. The
forests are dominated by Dahurian larch (*Larix gmelinii* Rupr.) and Scots pine (*Pinus sylvestris* L. var.
*mongolica* Litv.). Soils are predominantly brown coniferous and dark-brown forest peat (Xu, 1998).
Meteorological data was collected from stations nearest our sample sites (Xiaoergou station; Table S1).
The annual mean temperature ranges from -2.6 to 2.0 °C. The coldest and hottest month is January (-
39.5 °C) and July (32.8 °C), respectively. Annual precipitation ranges from 289 to 1000 mm (average
500 mm) with high interannual variations. Rain from June to August accounts for 68% of total annual
precipitation (Fig. 2). The relative humidity is low except for the growing season. Severe drought
occurs frequently, especially in spring and early summer (Sun, 2007), and leads to high fire risk. This
region has the highest average annual burned area in China (Sun, 2007).
**2.2 Tree-ring data**
Tree-ring cores were sampled from four Scots pine-dominated sites that are rarely disturbed in the
central Daxing'an Mountains in May 2011 and 2012. Each sampled tree was selected to avoid the
influence of identifiable stand disturbances (including animal and human disturbance, windstorm, snow,
and fire damage) and any obvious abnormal growth. The distance between sample sites is more than
100 km (Fig. 1). A total of 120 cores were obtained from living old trees at breast height (*ca*. 1.3 m)
(Table 1) using a 5.15-mm-diameter increment borer (500 mm length, two screws, Haglöf Sweden,
Längsele, Sweden). All cores were dried, mounted, surfaced, and cross-dated following standard
techniques of dendrochronology (Cook and Kairiukstis, 1990; Fritts, 1976). Ring widths were measured
with a precision of 0.001 mm using a Velmex measuring system (Velmex, Inc., Bloomfield, NY, USA).
The quality of cross-dating and measurement was evaluated using the COFECHA program
(Holmes, 1983). Two cores with weak correlation to the master chronology were excluded from further
analysis. Successively, the age-related trends were removed by fitting a cubic smoothing spline with a
50% frequency response cut-off at 2/3 of the series length using the ARSTAN program (Cook and
Kairiukstis, 1990). Tree-ring index was calculated as the ratio of the observed value to the estimated

growth curves. Autocorrelation was removed by autoregressive modelling, and the site chronology was calculated using a bi-weighted robust mean (Cook and Kairiukstis, 1990).

Four chronologies have high values of standard deviation, mean sensitivity, mean series correlation and agreement within population. They reflect high inter-annual variation and a strong common signal and are excellent proxies for regional climate. Since the four chronologies fit well (Table 2), we merged all samples to develop a single robust regional chronology (Fig. S1). Running RBAR (mean correlation between series) and EPS (expressed population signal) statistics were calculated using a 51-year interval of the chronology with a 25-year overlap to assess confidence in the chronology. RBAR averages variance among ring width series in a chronology, which estimates chronology signal strength (Cook and Kairiukstis, 1990). EPS estimates the degree to which the chronology represents a hypothetical chronology based on a finite number of trees that match a hypothetically perfect chronology; EPS values greater than 0.85 are generally considered acceptable threshold for a reliable chronology (Wigley et al., 1984). The regional chronology spanned the period from 1725 to 2010, and the reliable interval (EPS > 0.85) was 1751-2010 corresponding to eight trees (Fig. S1).

**2.3 Climate and statistical analyses**

Climate data were obtained from the National Meteorological Information Center (http://data.cma.cn/). The weather station nearest to the sample sites is Xiaoergou (Table S1 and Fig. 1), about 70-91 km away. Large-scale climate data (e.g. El Niño-Southern Oscillation, ENSO, Atlantic Multidecadal Oscillation, AMO; Pacific Decadal Oscillation, PDO; North Atlantic Oscillation, NAO) and high-resolution gridded climate data (Table S1; e.g. gridded temperature, precipitation and drought indices) were downloaded from the website: http://climexp.knmi.nl/. Pearson correlation analysis was

conducted to estimate climate-growth relationships. The gridded climate dataset is much longer and has higher homogeneity and coherency than station data (Fig. 2), the gridded monthly total precipitation (CRU GPCC; Schneider et al. (2015)) and mean temperature (CRU TS3.23; Jones and Harris (2013)) nearest to our sites were used for climate response analyses. In addition, the nearby gridded monthly Palmer Drought Severity Index (PDSI) data from Dai (2011) (Dai-PDSI, hereafter) was used to assess the effects of drought. Correlation analyses between the regional chronology and monthly climatic records were calculated from the previous July to the current July.

A linear regression model was used to reconstruct the drought variation, and a traditional split-period calibration and verification method was applied to examine the model fitness (Fritts, 1976). Statistical parameters included the $R^2$, Sign test (ST), reduction of error (RE), coefficient of efficiency (CE), product means test (PMT) and root mean square error (RMSE) (Cook and Kairiukstis, 1990; Fritts, 1976). Spatial correlation of the measured and reconstructed drought variables with regional gridded CRU-PDSI (Schrier et al., 2013) were performed to examine the spatial representativeness of our reconstruction using the KNMI climate explorer. Local historical drought data recorded in "Meteorological disasters dictionary of China" (Shen, 2008; Sun, 2007) were used to verify our PDSI reconstruction.

We also carried out the superposed epoch analysis (SEA) between the nearby forest fire events and the drought series to further validate the accuracy of our reconstruction because seasonal or annual droughts are usually a key factor of forest fire severity in the Daxing'an Mountains (Shen, 2008; Sun, 2007). Two regional forest fire chronologies (Mengkeshan and Pangu) reconstructed by tree-ring scars in nearby forests were used (Yao et al., 2017). The SEA was carried out using the software package

FHAES V2.0.0 (https://www.frames.gov/partner-sites/fhaes/download-fhaes/). In addition, the
consistency between our reconstruction and other local drought related time series including the gridded
Standardized Precipitation-Evapotranspiration Index (SPEI), Monsoon Asia Drought Atlas (MADA)
from Cook et al. (2010) (Cook-PDSI, hereafter) and Self-calibrating PDSI from Schrier et al. (2013)
(scPDSI, hereafter). We also compared our reconstruction with the nearby tree-ring-based hydroclimatic
reconstructions (the December-March precipitation reconstruction of the A'li River (AR) in the
Daxing'an Mountains from Lv and Wang (2014), the April-August SPEI reconstruction of the Hulun
Buir steppe (HB) on the east edge of Mongolian Plateaus in the western Daxing'an Mountains (Bao et
al., 2015), and the tree-ring-based streamflow reconstruction of Selenge River (SR) from Davi et al.
(2006) in the Mongolian Plateaus, Mongolia) to assess the reliability of reconstruction by filtering and
moving correlations.
To identify spatiotemporal patterns of drought variation in Northeast Asia and their relationship
with our reconstructed drought series, we analyzed the correlations between our series and other four
hydroclimatic reconstruction series in the Daxing'an Mountains and the Mongolian Plateau (Fig. 1). To
better visualize the comparison all series described above were standardized using Z-scores and
smoothed with a 21-year moving averaged to highlight low-frequency drought signals.
To evaluate the extreme dry and wet years in the historical period, we defined extreme dry and wet
years with the annual PDSI value being lower or higher than average +/- 1.5 standard deviation. We
assessed the multiyear dry/wet periods based on the intensity (average departure values from the long-
term mean) and magnitude (cumulative departure values from the long-term mean).
A spectral analysis was applied to identify the periodicity of dry/wet variation and possible effects
of large-scale climate using the Multi-taper method (MTM) (Mann and Lees, 1996). To further confirm
the linkage between large-scale climate and regional drought, we analyzed their relationship with
Pearson correlation analysis. Teleconnections between the reconstructed drought series and the global
sea surface temperature ($0.5° \times 0.5°$) were carried out to verify the potential drivers of large-scale
climate, such as ENSO, PDO and AMO, on local drought. To explore the linkages between the
reconstructed Dai-PDSI extreme events and atmospheric circulation patterns in Asia, the NCEP climate
data (Kalnay et al., 1996) were used to create January-December composite anomaly maps of the SSTs,
the 200-hPa geopotential height and vector wind in the wettest 10 years and driest 10 years during the
period 1948-2010.

**3 Results**
**3.1 Tree growth-climate relationships**

The radial growth of Scots pine was significantly ($p < 0.05$) positively correlated with precipitation

in all months except the previous November and current February (Fig. 3a) and temperature from the
previous November to current May (except for the current April) (Fig. 3a). The highest Pearson's
correlation coefficients occurred in October precipitation ($R = 0.35$, $p < 0.05$) and the previous
December mean temperature ($R = 0.35$, $p < 0.05$). Radial growth of Scots pine in the Daxing'an
Mountains was influenced by both precipitation and temperature simultaneously, but the effects of
precipitation were stronger, revealing the annual precipitation sensitivity of Scots pine during the last
century (Fig. 3a). Furthermore, we calculated the correlation between the tree-ring index and Dai-PDSI
(common period of 1901-2010), which takes into account temperature and precipitation (Dai, 2011).
Significant ($p < 0.05$) positive correlations between tree rings and PDSI was found in all months from
the previous July to the current July (Fig. 3b). The correlation between tree growth and annual (Jan-
Dec) average PDSI showed the highest correlation coefficient ($R = 0.62$, $p < 0.0001$, $n = 110$) among all
seasonal PDSI compositions. The results confirmed that water availability had a significant limiting
influence on Scots pine growth in the last century (Fig. 3).

**3.2 PDSI reconstruction**

The linear model for PDSI reconstruction is:

$D_t = 6.69 \, I_t - 7.13$, ($R = 0.62$, $N = 100$, $F = 60.52$, $p < 0.0001$)         (1)

where $D_t$ is the annual PDSI and $I_t$ is the tree-ring index at year $t$. The split calibration-verification
test showed that the explained variances were high during the two calibration periods (1911-1960 and
1961-2010). The statistics of $R$, $R^2$, ST, PMT are all significant at $p < 0.05$, which indicated that the
model was reliable (Table 3). The most rigorous tests, RE and CE, were also positive for both
verification periods (Cook and Kairiukstis, 1990; Fritts, 1976) (Table 3). For the calibration period
1911-2010, the reconstruction explained 38.2% of the PDSI variation (37.6% after accounting for the
loss of degrees of freedom). These results suggest that the linear model is robust for PDSI
reconstruction.
The instrumental and reconstructed PDSI of Central Daxing'an Mountains have similar trends and
are parallel to each other during the calibration period (Fig. 4). However, the reconstructed PDSI did not
capture the magnitude of extreme dry or wet conditions. Spatial correlation analysis showed that the
instrumental and reconstructed PDSI had a strong and similar spatial correlation pattern with the
Northeast Asia gridded Dai-PDSI (Fig. 5).
**3.3 Historical PDSI variability**
The reconstructed annual PDSI with an 11-year moving average exhibited a mean of 0.48 and a
standard deviation (SD) of ±1.15 during the past 260 years (Fig. 4b). Reconstruction of the annual PDSI
displayed strong interannual to decadal scale variability throughout the period 1751-2010. During the
last 260 years, there were 22 extreme dry years (accounting for 8.5%) and 15 extreme wet years (5.8%)
(Table 4). Most extreme dry years occurred in the 19th (12 years, accounting for 48%) and 20th (9 years,
accounting for 36%) centuries, and most extreme wet years occurred in the 20th century (9 years,
accounting for 60%). Among the extreme years, 1784, 1853, 1818, 1862 and 1863 were the five driest
years, and 1998, 1952, 1770, 1993 and 1766 were the five wettest years (Table 4). We also found that
many extreme dry or wet years occurred in succession, for example 1862 and 1863.
Compared with the severe single-year droughts, multi-year droughts had a greater effect on tree
growth, and we defined the dry and wet periods as those when the 11-year moving average PDSI was
more than 0.5 SD from the mean for at least 2 consecutive years. Four dry periods, AD 1751-1752,
1812-1817, 1847-1866 and 1908-1927, and four wet periods 1757-1771, 1881-1902, 1952-1955 and
1989-2004 were identified (Table 5). The dry periods of 1847-1866 and 1906-1927 were the longest,
spanning 20 years, while the longest wet period, from 1881-1902, lasted for 22 years (Table 5). The
multiyear drought in 1847-1866 was the most serious due to the long duration and intensity, and the
period 1906-1927 was the second most significant drought (Table 5). Wet periods in 1757-1771 and
1989-2004 were the most remarkable in terms of their intensity and duration (Table 5).
Spectral analysis revealed that the historical PDSI variation in the Daxing'an Mountains showed
several significant (95% or 99% confidence level) periodicities at 46.5-78.7 (99%), 12, 5-6 (99%), and
2-3 (99%) years, which corresponded to significant cycle peaks presented in Fig. 6.

## 4 Discussion

### 4.1 Climate-growth relationship

Scots pine is an drought-tolerant species and drought stress is thought to be the main climate factor
limiting its radial growth in semi-arid or semi-humid regions, such as in the Mongolian Plateaus and
western Daxing'an Mountains (Bao et al., 2015; Davi et al., 2006; Liu et al., 2009; Pederson et al.,
2013). Previous dendroclimatic studies from these regions suggest that radial growth of Scots pine is
sensitive to humidity, precipitation or drought (e.g. PDSI, SPEI), and most analyses have reconstructed
hydroclimatic history (Bao et al., 2015; Liu et al., 2009). In these areas, the radial growth of Scots pine
usually has a typical climate (drought) response pattern with positive tree growth response to increasing
precipitation and a negative response to increasing temperature (Bao et al., 2015; Davi et al., 2006; Liu
et al., 2009). This typical drought response pattern is usually found in other drought or wetness tree ring
reconstructions (Li et al., 2016; Liu et al., 2017). In this study, the correlation between tree-ring index
and monthly precipitation and temperature revealed that the radial growth of Scots pine was mainly
limited by water, which is consistent with the physiological characteristics of tree species living in
semi-arid regions. A significant positive relationship between the tree-ring index and PDSI in all
months supported moisture as the main limiting factor of Scots pine radial growth (Fig. 3b).

The drought response was also found in Dahurian larch (Wang and Lv, 2012), another important conifer tree species in the study area. However, the typical drought response to temperature was not obvious, and the radial growth of Scots pine was not significantly negatively correlated with growing season (July-September) temperature (Fig. 3a). On the contrary, a significant positive response of radial growth to the non-growing season temperature was found. It is possible that higher winter temperatures could protect dormant buds from frost damage (Chen et al., 2012). The positive correlation with spring temperature could be due to earlier and larger snow melting, which supplies the spring soil water, and eventually stimulates tree growth (Hollesen et al., 2015; Zhu et al., 2017). This unusual drought response pattern might be due to the relatively humid climate and the northern latitude of our study sites, where the positive effect of temperature was greater than the negative effect resulting from drought stress (Wang and Song, 2011). Similar drought response patterns were also found in tree-ring-based drought reconstructions in the middle Qilian Mountains (Sun and Liu, 2013) and the Tienshan Mountains of western China (Chen et al., 2015).

**4.2 Comparison with regional records**

We used the local historical document records to verify our PDSI reconstruction for the timing of extreme dry years or periods. During the last 260 years, 60.1% (13/22) of extreme dry years were noted in historical documents (Shen, 2008; Sun, 2007). Tree rings cannot fully record the continuous drought events (years) resulting in a limited percentage or correspondence. For example, only 1861 was recorded in our reconstruction during the extreme drought period 1860-1865. Thus, some severe drought events affect radial tree growth in some but not all years (Fritts, 1976). In addition, the lag response of radial growth to climate (drought) might have a great contribution to unrecorded extreme

drought events (Fritts, 1976). For example, the local historical documents recorded the dry years of
1817 and 1855 that showed narrower rings or as an extreme dry event in the following year. Two
multiyear droughts recorded in tree-rings, 1847-1866 and 1908-1927, can both be identified in historical
documents (Shen, 2008; Sun, 2007). Moreover, the SEA between our reconstructed drought series and
forest wildfire history revealed that a significant drop of PDSI values occurred during the year of the
forest fire in Mengkeshan and Pangu (Fig. S2), further validating the accuracy of our reconstruction.
Spatial correlation analysis indicated a strong pattern between our reconstruction and gridded
scPDSI in Northeast Asia (Fig. 7), and our reconstruction also represented drought/wet variations in
surrounding geographic regions. During the common periods, our reconstruction shared a similar
dry/wet fluctuation with precipitation of the A'li River (Wang and Lv, 2012) and SPEI of Hulun Buir
steppe (Bao et al., 2015) both in the low and high frequency (Fig. 7b-d). Significant ($p < 0.05$)
correlations among them were found in low and high frequency and some common dry/wet periods,
highlighted in Fig. 7, which confirmed that our drought reconstruction could account for the most
dry/wet variations in the Daxing'an Mountains.
It's important to note that our drought reconstruction and the MADA by Cook et al. (2010) from the
same PDSI grid showed a complete opposite trend ($R_L = -0.19$; $p < 0.01$) in low frequency (Fig. 7).
Negative correlations between the MADA and the SPEI ($R_L = -0.31$; $p = 0.03$) and scPDSI ($R_L = -$
$0.126$; $p = 0.236$), positive correlations between our drought reconstruction and the SPEI ($R_L = 0.95$; $p$
$< 0.01$) and scPDSI ($R_L = 0.807$; $p < 0.01$) were also found, although it had a seasonal difference with
our drought reconstruction. These imply that the MADA by Cook et al. (2010) might be inaccurate or
even reverse in the timing of dry/wet variations in the Daxing'an Mountains. Similar divergence of tree-

ring-based drought reconstruction between the MADA and individual sampling sites was also found by Li et al. (2015) from Guancen Mountain and Liu et al. (2017) from central Inner Mongolia. The insufficient spatiotemporal distribution of tree-ring networks, especially in eastern China, used in MADA might be the main reason for this divergence/inaccuracy (Cook et al., 2010; Li et al., 2015; Liu et al., 2017). Therefore, our drought reconstruction is necessary to improve our understanding of the East Asian Monsoon climate variability.

On a larger spatial scale, the streamflow reconstruction of Selenge River in the West-Central Mongolian Plateaus from Davi et al. (2006) presented a significant positive correlation with our drought reconstruction in low frequency ($R_L = 0.29$; $p < 0.01$) during the full periods. Our reconstructed PDSI also displayed some common variation trends for dry/wet periods with the reconstructed streamflow variations from the Selenge River (Davi et al., 2006), especially at the decadal scale. These relationships suggest that there are common drivers affecting dry/wet variations of the Daxing'an Mountains and the West-Central Mongolian Plateau, although there might be some discordance. Among those differences, the most obvious one is the completely different dry/wet variation trends among the Daxing'an Mountains (wetter), the West-Central Mongolian Plateaus (mild drier) as well as their transition zones, the East Mongolian Plateaus (Hulun Buir steppe; drier) since the late 1970s (Fig. 8a). Similar results were also found by Dai (2013), which presented a different dry-wet pattern under global warming using observations and models. In the Tibetan Plateau, Li et al. (2016) found moisture increases related to rapid warming (warm-wet). Although the reason for this divergence should be further studied, it might be related to the different response to the phase shift (negative to positive) of the PDO in 1976 and 1977 (Ma, 2007; Wang et al., 2014). Ma (2007) found that the positive PDO phase usually corresponds to the

drought period with warming and less precipitation, while the negative PDO phase often matches the

wet period with low temperature and more precipitation. Simultaneously, the drought trend caused by

the persistent significant warming in semi-arid or semi-humid regions might be more serious than in

humid regions (Dai, 2013). In addition, a different record of severe drought that occurred over a large

geographic area in northern Asia during the period 1920s-1930s, has been reported by many other

studies in north China (Bao et al., 2015; Chen et al., 2015; Liang et al., 2006; Liu et al., 2009). As

indicated by the tree-ring series, the drought event during the period 1920s-1930s in the Daxing'an

Mountains was more severe than on the East Mongolian Plateaus (the Hulun Buir steppe), which was

consistent with the result by Dong et al. (2013). The drought, however, was not found in the West-

Central Mongolian plateaus (the Selenge River). On the contrary, it was very wet at that time (Fig. 8).

Different spatial patterns of severe drought over northeast Asian might be associated with the intensity

and scope of the strong ENSO during this period (Dong et al., 2013).

**4.3 Linkages to the Pacific and Atlantic Oceans**

Spectral analysis revealed that several significant cycles existed in our drought series (Fig. 6). The

significant high-frequency 2.0−5.8-year periodicities were within the 2−7 year cycles of ENSO (Li et

al., 2013), so the drought variations in the Daxing'an Mountains might be related to ENSO. Similarly,

the local dry-wet changes related to ENSO has been confirmed by other tree-ring-based hydroclimatic

reconstructions in northeast China (Bao et al., 2015; Lv and Wang, 2014; Wang and Lv, 2012),

northwest China (Chen et al., 2015; Sun and Liu, 2013) and the Mongolian Plateaus (Davi et al., 2006).

A strong connection appears between our reconstruction and annual SSTs over the Pacific Ocean,

especially nearby the equator, the north Pacific, as well as the east and west coasts of the Pacific Ocean

(Fig. S3). The significant positive correlation between the Niño 3 index and the dry-wet index   in both
low and high frequencies (Table 6, Fig. 8b). also confirmed the potential links between ENSO and the
dry/wet variations in the Daxing'an Mountains. Although the mechanisms need to be further studied,
the close relationship between the oscillatory changes of North Atlantic SST and the Asian monsoon
have been demonstrated (Zuo et al., 2013). ENSO might indirectly influence dry-wet changes in the
Daxing'an Mountains by affecting the local climate (Shuai et al., 2016). Wang et al. (2013) found that
the ENSO could potentially drive or affect the Asian monsoon, which in turn affects temperature and
precipitation to drive local drought variations, as a possible driving mechanism (Fig. 9). Significant
positive correlations between the Niño 3 index and local climate (temperature and precipitation) further
confirms our inference (Table 6).
The 12-year cycle indicated that dry-wet changes in the Daxing'an Mountains might be influenced
by solar activity (Shindell et al., 1999). Several previous studies have demonstrated that solar activity
can influence the local dry-wet variations (Chen et al., 2015; Hodell et al., 2001; Sun and Liu, 2013). In
northeastern China, Hong et al. (2001) also found the signals of solar activity in a 6000-year record of
drought and precipitation. Significant positive correlations between the Total Solar Irradiance (TSI;
reconstruction from IPCC AR5) and the dry-wet index in the Daxing'an Mountains in low and high
frequencies, and between the TSI and the local climate (temperature and precipitation) further
confirmed a possible relationship between solar activity and local drought (Table 6, Fig. 8b). Wang et
al. (2005) found a potential link between the Asian monsoon and solar changes. Dry-wet changes in the
Daxing'an Mountains might be driven by the Asian monsoon which is influenced by solar activities
(Fig. 9).
Cycles of 46.5 - 48.8 years might be related to the PDO since it coincided with the 50-70 year cycle
of PDO (Macdonald and Case, 2005). This was verified by the strong connection between our drought
reconstruction and annual SSTs over the Pacific Ocean (Fig. S3). The cycles/signals of PDO widely
exist in most tree-ring-based drought reconstructions (Bao et al., 2015; Chen et al., 2015; Sun and Liu,
2013; Wang and Lv, 2012), and many studies have confirmed that PDO can influence drought
conditions in China (Bao et al., 2015; Cook et al., 2010; Ma, 2007). The potential linkages between the
PDO and local drought in the Daxing'an Mountains is further confirmed by the significant positive
correlations between the PDO index (Mann and Lees, 1996) and the dry-wet index in low and high
frequencies (Table 6, Fig. 8b). The positive/warm phase of PDO usually corresponds to the dry period,
while the negative/cold phase corresponds to the wetting period (Ma, 2007). For example, the severe
drought in 1920s-1930s corresponds to the PDO negative phase. Significant positive correlations
between the PDO index and local climate (Fig. 9) suggest that the PDO might affect the dry-wet
changes in the Daxing'an Mountains by regulating the intensity or location of the Asian monsoon (Bao
et al., 2015; Cook et al., 2010; Ma, 2007). Similar results were found in a nearby tree-ring-based
drought reconstruction (Bao et al., 2015).
The 73-years drought cycle might be derived from oscillatory changes in the North Atlantic SST-
(Knudsen et al., 2011). Spatial correlations between our drought series and annual SSTs also show a
strong teleconnection across the Atlantic Ocean (Fig. S3), which further confirmed potential linkages
between the North Atlantic SSTs and dry-wet changes in the Daxing'an Mountains. Although our
research area is far from the Atlantic, some studies have confirmed that large-scale climate oscillations
in the Atlantic Ocean (such as the Atlantic Multidecadal Oscillation (AMO), North Atlantic Oscillation
(NAO) as well as Summer NAO (SNAO)) could affect local climate and tree growth in China (Bates,
2007; Linderholm et al., 2011; Linderholm et al., 2013; Sun et al., 2008; Wang et al., 2011). Most tree-
ring drought reconstructions also found the signals of oscillatory changes correlated with the North
Atlantic SSTs (e.g. AMO, NAO and SNAO), such as in the Daxing'an Mountains (Lv and Wang, 2014;
Wang and Lv, 2012), eastern Mongolian Plateaus (Bao et al., 2015; Liu et al., 2009), West-Central
Mongolia (Davi et al., 2006), and northwest China (Chen et al., 2015; Sun and Liu, 2013). Furthermore,
we also identified a significant negative/positive correlation between the dry-wet change in the
Daxing'an Mountains and the AMO, NAO and SNAO index both in low or high frequency (Table 6,
Fig. 8c). The strong AMO signal (Wang et al., 2011) and teleconnections with SNAO (Linderholm et
al., 2013) also have been found in tree-ring widths of Scots pine in northeast China and east central
Siberia during the last 400 years. These studies all confirmed that oscillatory changes in the North
Atlantic SST (e.g. AMO, NAO and SNAO) could drive dry-wet changes in the Daxing'an Mountains.
Although its mechanism needs to be further studied, the close relationship between the oscillatory
changes in the North Atlantic SST and the Asian monsoon has been demonstrated. Recent studies have
shown that the AMO (Wang et al., 2013), NAO (Feng and Hu, 2008) and SNAO (Linderholm et al.,
2011) all could drive or affect the Asian monsoon. In this study, although only the AMO index was
significantly correlated with local climate (Table 6, Fig. 8c), it also confirmed that the oscillatory
changes in the North Atlantic SST, especially the AMO, could drive wet-dry changes in the Daxing'an
Mountains by influencing the Asia Monsoon (Bao et al, 2015; Chen et al., 2015; Cook et al., 2010; Li et
al., 2015; Linderholm et al., 2011; Sun et al., 2008).

Previous studies have found that drought variation in northeast Asia may be associated with Asian monsoon activity (Bao et al., 2015; Chen et al., 2015; Cook et al., 2010; Li et al., 2015; Linderholm et al., 2011; Sun et al., 2008). In wet years, the strengthened southerlies and easterlies entered inland China associated with a positive pattern over northeast Asia and some negative height-anomaly centers in west Russia and south Asia as well as the Indian and north Pacific oceans, which strengthened the westerly circulation (Fig. 10a, c). In dry years, however, strengthened southerlies and south-westerlies entered northeast China associated with a positive pattern over east Asia and western Russia, and some negative height-anomaly centers in southern Russia and south Asia as well as the Indian and south Pacific oceans (Fig. 10a, c).

The composite of 200-hPa geopotential height of the most humid 10 years (positive anomaly) in the central-north Daxing'an Mountains is opposite to that of the most arid 10 years (negative anomaly) (Fig. 10c, d). Positive and negative SST anomalies were also found in the western and northern Pacific Ocean during the wettest and driest years (Fig. 10e, f). In the wet years, abundant moisture is transported from the Pacific Ocean through Mongolian Plateau to the Daxing'an Mountains *via* the strong east Asian monsoon's southeasterly moisture flux joined with a strong Westerly circulation (Fig. 10a). This negative anomaly combined with positive SST in the western and northern Pacific Ocean lead to an enhanced dry jet (south-westerlies) across/toward the Daxing'an Mountains (Fig. 10b, c, e). Several studies have reported that the dry and wet variations in northeast Asia are strongly linked with the Asian monsoon and SSTs in the Pacific and Atlantic oceans (Bao et al., 2015; Chen et al., 2015; Cook et al., 2010; Li et al., 2015; Linderholm et al., 2011). In addition, the potential evaporation pattern

in the Daxing'an Mountains is extremely low in the wettest years, and it also supports the above
remote-connection assumptions (Fig. S4).

**5 Conclusion**

We developed a 260-year (1751 to 2010) tree-ring chronology of Scots pine (*Pinus sylvestris* L.
var. *mongolica* Litv.) from four sample sites in the central Daxing'an Mountains, northeast China.
Radial growth of Scots pine was mainly limited by water availability ($R = 0.62$, $p < 0.01$). A 260-year
dry-wet change history was reconstructed, and the reconstruction equation explained 38.2 % of the
PDSI variance for the period 1911-2010. Four dry and wet periods were found in the past 260 years,
respectively. The extreme dry years in our reconstruction series are consistent with the local historical
document records. Our reconstruction series revealed the dry-wet changes in the Daxing'an Mountains,
and also was the representative of the dry-wet changes in the West-Central Mongolian Plateaus,
especially at the decadal scale. Droughts during the 1920s-1930s in the Daxing'an Mountains were
more severe than in surrounding areas. In addition, the reconstruction series showed that the Daxing'an
Mountains is getting warm and wet since the late 1970s. This is not in line with the situation in the
Mongolian Plateaus, especially in the transition zones. Our reconstruction also suggests that the MADA
by Cook et al. (2010) may not be accurate in the Daxing'an Mountains likely due to the insufficient
spatiotemporal distribution of the tree-ring data in this area. Overall, drought variability in the central
Daxing'an Mountains and its relationship with the surrounding areas might be driven by climate
oscillations of the Pacific and Atlantic Oceans (e.g., ENSO, PDO, AMO, NAO and SNAO). These
large-scale climate oscillations affect the Asian monsoon and then lead to dry and wet changes in
Daxing'an Mountains.
**Acknowledgements** This research was supported by the Key Project of the China National Key
Research and Development Program (2016YFA0600800), the National Natural Science Foundation of
China (Nos. 41471168 and 31770490), the Program for Changjiang Scholars and Innovative Research
Team in University (IRT-15R09), and the Fundamental Research Funds for the Central Universities
(2572016AA32). We also thank Yongxian Lu and Lei Zhang of Northeast Forestry University for their
assistance in the field.

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

**Table 1** Site description and statistical characteristic of the *Pinus sylvestris* var. *mongolica*
dendrochronologies in the Daxing'an Mountains.

| Site | Latitude (N) | Longitude (E) | Elevation (m) | Number of trees | Time span | EPS[a] | RBAR[b] |
|---|---|---|---|---|---|---|---|
| Keyihe (KY) | 50°39′44.8″ | 122°23′13.3″ | 550 | 36 | 1725-2010 | 0.93 | 0.57 |
| Alihe (AL) | 50°38′37.7″ | 124°28′28.7″ | 380 | 32 | 1742-2010 | 0.87 | 0.52 |
| Ganhe (GH) | 50°43′51.9″ | 123°05′56.9″ | 760 | 19 | 1793-2010 | 0.88 | 0.59 |
| Jinhe (JH) | 50°26′16.7″ | 121°59′46.7″ | 830 | 33 | 1769-2011 | 0.95 | 0.61 |
| Region (RE) | - | - | - | - | 1725-2010 | 0.97 | 0.51 |

[a] Expressed population signal statistic.
[b] RBAR = the mean correlation coefficient between all tree-ring series used in a chronology.
**Table 2** Five-chronology correlation matrix over the common period 1793-2010.

|     | AL      | GH      | JH      | Region  |
| --- | ------- | ------- | ------- | ------- |
| KY  | 0.38**  | 0.46**  | 0.55**  | 0.81**  |
| AL  |         | 0.33**  | 0.32**  | 0.68**  |
| GH  |         |         | 0.32**  | 0.72**  |
| JH  |         |         |         | 0.74**  |

** Significance level ($p < 0.01$). The site codes are identical with those in Table 1.
**Table 3** Calibration and verification statistics of the PDSI reconstruction

| Calibration | $R$ | Verification | $R^2$ | RE | CE | ST | PMT | RMSE |
|---|---|---|---|---|---|---|---|---|
| 1911-2010 | 0.62** | - | - | 0.38 | - | (74, 26)** | 8.04** | 1.4 |
| 1961-2010 | 0.53** | 1911-1960 | 0.47** | 0.47 | 0.47 | (39, 11)** | 5.24** | 1.34 |
| 1911-1960 | 0.69** | 1961-2010 | 0.28** | 0.28 | 0.25 | (34, 16)* | 6.23** | 1.25 |

* = $p < 0.05$, ** = $p < 0.01$. RE-reduction of error, CE-coefficient of efficiency, ST-sign test, PMT-
product means test, RMSE-root mean square error.

**Table 4** Reconstructed extreme dry/wet years and annual PDSI of the Daxing'an Mountains.

| Dry year (Rank) | PDSI | Dry year (Rank) | PDSI | Wet year (Rank) | PDSI |
|---|---|---|---|---|---|
| 1784 (1) | -3.574 | 1909 (16) | -2.484 | 1998 (1) | 2.521 |
| 1853 (2) | -3.315 | 1916 (17) | -2.479 | 1952 (2) | 2.091 |
| 1818 (3) | -3.238 | 1854 (18) | -2.405 | 1770 (3) | 2.020 |
| 1862 (4) | -3.006 | 1865 (19) | -2.314 | 1993 (4) | 2.011 |
| 1863 (5) | -3.001 | 1861 (20) | -2.310 | 1766 (5) | 1.790 |
| 1918 (6) | -2.991 | 1864 (21) | -2.283 | 1897 (6) | 1.728 |
| 1919 (7) | -2.977 | 1856 (22) | -2.275 | 1996 (7) | 1.663 |
| 1915 (8) | -2.882 | | | 1899 (8) | 1.655 |
| 1917 (9) | -2.777 | | | 1755 (9) | 1.576 |
| 1852 (10) | -2.733 | | | 1999 (10) | 1.548 |
| 1851 (11) | -2.716 | | | 2000 (11) | 1.488 |
| 1860 (12) | -2.695 | | | 1997 (12) | 1.451 |
| 1967 (13) | -2.671 | | | 1994 (13) | 1.422 |
| 1925 (14) | -2.660 | | | 1769 (14) | 1.405 |
| 1911 (15) | -2.581 | | | 1764 (15) | 1.279 |



**Table 5** The long-term droughts and pluvial in the Central Daxing'an Mountains during the last 260
years.

| Year | Dry/Wet | Duration (year) | Magnitude (sum of PDSI) | Intensity (mean PDSI) |
|------|---------|-----------------|-------------------------|------------------------|
| 1751-1752 | Dry | 2 | -2.36 | -1.33 |
| 1757-1771 | Wet | 15 | 19.52 | 1.30 |
| 1812-1817 | Dry | 6 | -3.73 | -0.62 |
| 1847-1866 | Dry | 20 | -32.70 | -1.64 |
| 1881-1902 | Wet | 22 | 18.98 | 0.86 |
| 1906-1927 | Dry | 20 | -31.79 | -1.59 |
| 1952-1955 | Wet | 4 | 2.78 | 0.69 |
| 1989-2004 | Wet | 16 | 19.67 | 1.23 |


**Table 6** Correlation coefficients between the large-scale climate index (AMO, PDO, NAO, SNAO, TSI, and Niño 3) and the local annual mean temperature, total precipitation, instrumental Dai-PDSI as well as the Z-score of dry-wet variation in the Daxing'an Mountains (DM$_{Z-score}$)

| | Temperature | | | Precipitation | | | Dai-PDSI | | | DM$_{Z-score}$ | | |
|---|---|---|---|---|---|---|---|---|---|---|---|---|
| | $R$ | $p$ | N | $R$ | $p$ | N | $R$ | $p$ | N | $R$ | $p$ | N |
| AMO | 0.44** | 0.00 | 106 | 0.30** | 0.00 | 106 | 0.44** | 0.00 | 96 | 0.35** | 0.00 | 282 |
| PDO | 0.46** | 0.00 | 106 | 0.39** | 0.00 | 106 | 0.51** | 0.00 | 96 | 0.34** | 0.00 | 282 |
| NAO | 0.17 | 0.08 | 106 | -0.04 | 0.71 | 106 | -0.08 | 0.43 | 96 | -0.21** | 0.00 | 282 |
| SNAO | 0.22* | 0.02 | 110 | 0.08 | 0.39 | 110 | 0.08 | 0.42 | 100 | 0.13* | 0.05 | 246 |
| TSI | 0.23* | 0.01 | 110 | 0.35** | 0.00 | 110 | 0.34** | 0.00 | 100 | 0.12* | 0.04 | 286 |
| Niño 3 | 0.34** | 0.00 | 106 | 0.26** | 0.01 | 106 | 0.28** | 0.01 | 96 | 0.14** | 0.02 | 282 |

Note: The AMO, PDO, NAO, SNAO, TSI and Niño 3 is the Atlantic Multidecadal Oscillation reconstruction from Mann et al. (2009), the Pacific Decadal Oscillation reconstruction from Mann et al. (2009), the Multi-decadal Winter North Atlantic Oscillation reconstruction from Trouet et al. (2009), the summer NAO based on the 20C reanalysis sea-level pressure reconstruction (SNAO), the Total Solar Irradiance reconstruction from IPCC AR5 and the Niño 3 reconstruction from Mann et al. (2009), respectively. All above data were downloaded from http://climexp.knmi.nl/. * $p < 0.05$, ** $p < 0.01$

**Figures captions**

**Fig. 1** Sampling sites and weather station distribution map. The red circles, start and black square are the sampled sites, PDSI grid point and the weather station, respectively. The red box represents the northern Daxing'an Mountains (this study). The blue and green box represents the east and west-central Mongolian Plateau, respectively.

**Fig. 2** Monthly total precipitation (P) and mean temperature (T) at the Xiaoergou (a) meteorological station (1957-2014) and grids (b) data (1901-2014); the annual total precipitation (c), and the annual mean temperature (d) and PDSI (e). The dashed line indicates the linear fitting values.

**Fig. 3** Pearson correlation coefficients between tree-ring index of *Pinus sylvestris* var. *mongolica* and the monthly total precipitation, mean temperature (a) and Dai-PDSI (b). Significant correlations ($p < 0.05$) are indicated by above or below the 95% confidence line (dash line). The minus sign "-" in abscissa represents the previous year, for example, "-7" represents the previous July.

**Fig. 4** The reconstruction PDSI series in the Daxing'an Mountains, northeast China. (a) Comparison of the observed (pink line) and reconstructed (red line) annual PDSI during the calibration period 1911-2010; (b) The reconstruction series of annual PDSI, plotted annually from 1725 to 2010 (green line), along with a smoothed 11-year moving average (red bold line); Blue filled triangles indicate a forest fire in nearby area reconstructed from tree-ring fire scars in Mengkeshan (higher) and Pangu (lower), northern Daxing'an Mountains.

**Fig. 5** Spatial correlation fields between (a) the instrumental and (b) reconstructed annual Dai-PDSI for the Daxing'an Mountains and the regional Dai-PDSI during the period 1911-2010 (http://climexp.knmi.nl/). The blue circle is the reconstructed PDSI grid.

**Fig. 6** Multi-taper method power spectrum of the reconstructed PDSI during the period 1751-2010. The 95% and 99% confidence level relative to red noise are shown and the numbers refer to the significant period in years.

**Fig. 7** Comparisons of (a) the drought reconstruction derived from the Monsoon Asia Drought Atlas (MADA, Cook et al. (2010)), (b) the winter precipitation reconstruction of the A'li River in northeastern China (AR, Lv and Wang (2014)), (c) the mean annual PDSI reconstruction in the Daxing'an Mountains (TS, in this study), (d) the April-August SPEI reconstruction of the Hulun Buir steppe in eastern Mongolian Plateaus (HB, Bao et al. (2015)) and (e) the April-October streamflow reconstruction of the Selenge River in northeastern Mongolia(SR, Davi et al. (2006)). All above series were standardized using Z-scores (high frequency) and then smoothed with a 21-year moving average (low frequency; red bold line). Blue and red shade areas represent a consistent period of drought and wetness, respectively. Correlation coefficients between our reconstruction series and other series in low ($R_L$) and high ($R_H$) frequency are shown on the diagram. ** $p < 0.01$

**Fig. 8** Comparisons of the drought reconstruction and other large-scale climate oscillations. (a) the dry-wet changes in the Daxing'an Mountains (DM, the average of our reconstruction and the precipitation reconstruction of the A'li River), the Mongolian Plateaus (MP, the streamflow reconstruction of the Selenge River) as well as their transition zones (TZ, the SPEI reconstruction of the Hulun Buir steppe); (b) the drought reconstruction in the Daxing'an Mountains (DM), the Pacific Decadal Oscillation (PDO) and the Niño 3 index reconstruction from Mann et al. (2009) (Nino3) as well as the Total Solar Irradiance reconstruction from IPCC AR5 (TSI); (c) the drought

reconstruction in the Daxing'an Mountains (DM), the Atlantic Multidecadal Oscillation

reconstruction from Mann et al. (2009) (AMO), the Multi-decadal Winter North Atlantic

Oscillation reconstruction by Trouet et al. (2009)(NAO) and the summer NAO based on the 20C

reanalysis sea-level pressure reconstruction (SNAO). All above series were standardized using Z-

scores and then smoothed with a 21-year moving averaged to highlight low-frequency drought

signals. Significant correlation coefficients (** $p < 0.01$) are listed in the figure.

**Fig. 9** Spatial correlations between the annual East Asian monsoon index and the local (a) temperature,

(b) precipitation and (c) scPDSI from AD 1948 to 2010.

**Fig. 10** Composite anomaly maps of the 200-hPa vector wind and geopotential height, and the SSTs

(from January to December) for the 10 wettest (a, b and e) and 10 driest (c, d and f) years of the

Dai-PDSI reconstruction during the period 1948-2010.



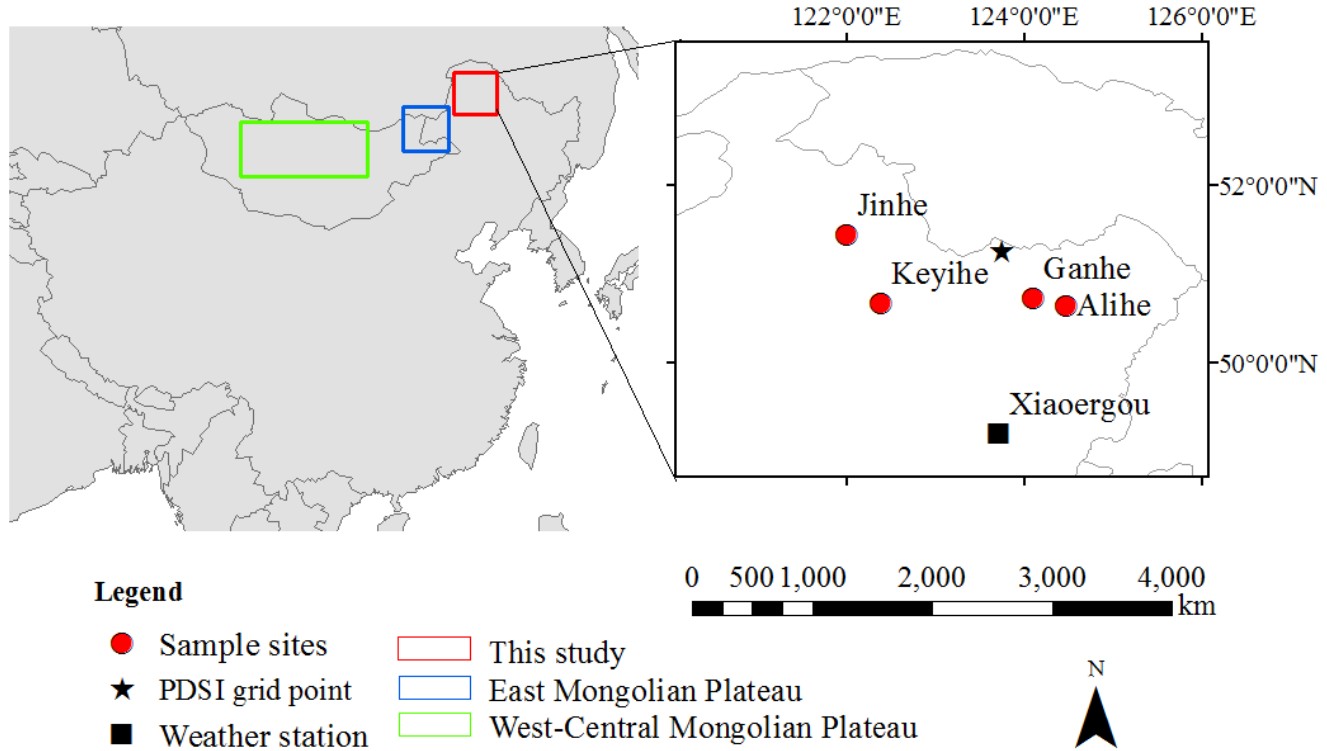


**Fig. 1** Sampling sites and weather station distribution map. The red circles, start and black square are the
sampled sites, PDSI grid point and the weather station, respectively. The red box represents the
northern Daxing'an Mountains (this study). The blue and green box represents the east and west-
central Mongolian Plateau, respectively.

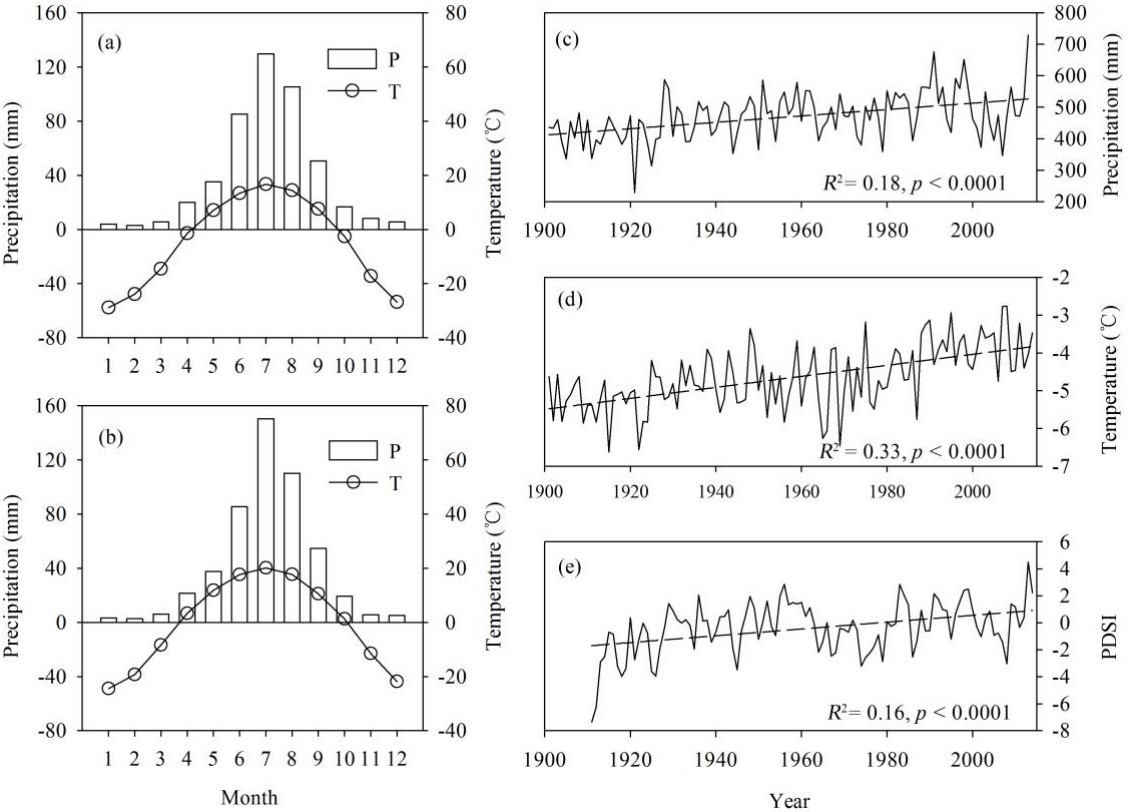


**Fig. 2** Monthly total precipitation (P) and mean temperature (T) at the Xiaoergou (a) meteorological
station (1957-2014) and grids (b) data (1901-2014); the annual total precipitation (c), and the annual mean
temperature (d) and PDSI (e). The dashed line indicates the linear fitting values.

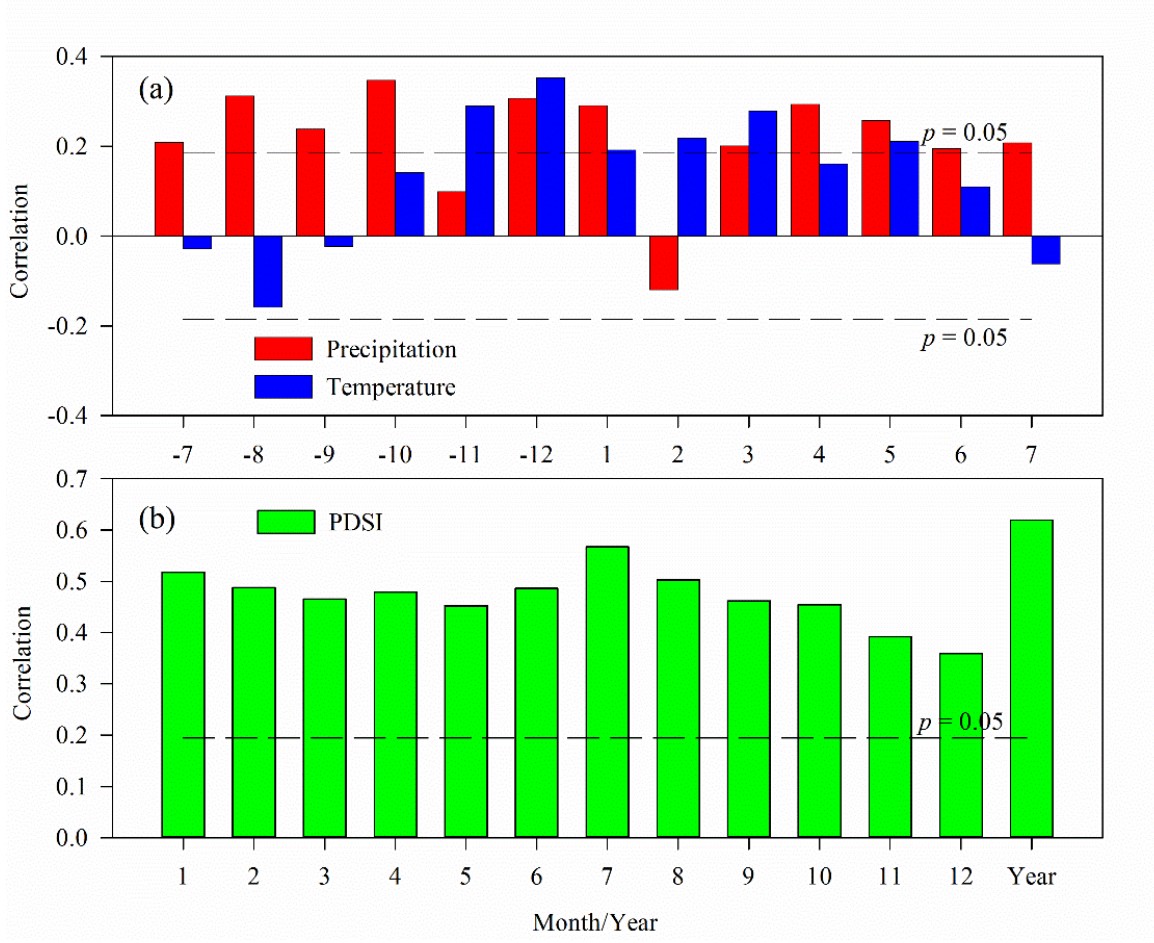


**Fig. 3** Pearson correlation coefficients between tree-ring index of *Pinus sylvestris* var. *mongolica* and
the monthly total precipitation, mean temperature (a) and Dai-PDSI (b). Significant correlations ($p <$
0.05) are indicated by above or below the 95% confidence line (dash line). The minus sign "-" in
abscissa represents the previous year, for example, "-7" represents the previous July.

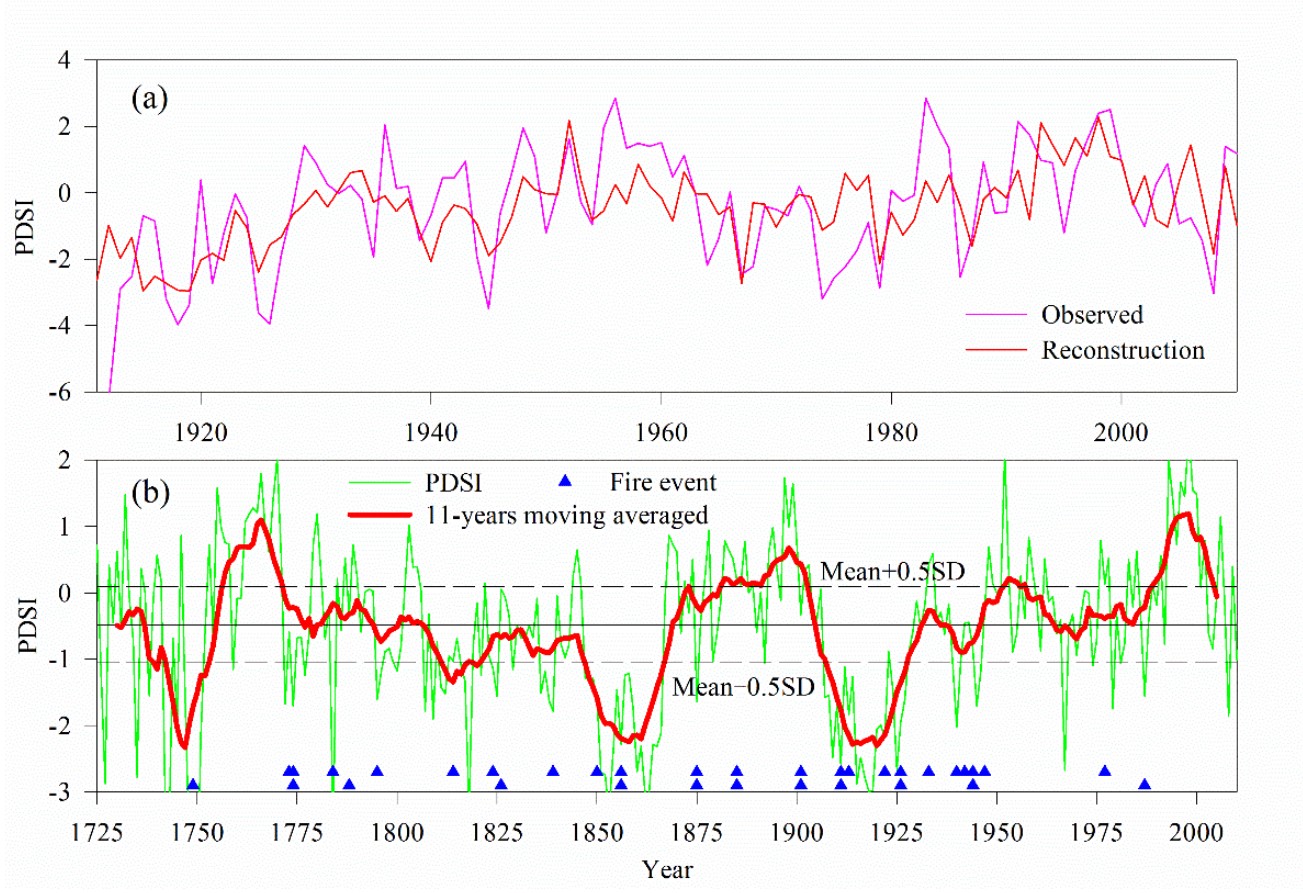


**Fig. 4** The reconstruction PDSI series in the Daxing'an Mountains, northeast China. (a) Comparison of

the observed (pink line) and reconstructed (red line) annual PDSI during the calibration period 1911-

2010; (b) The reconstruction series of annual PDSI, plotted annually from 1725 to 2010 (green line),

along with a smoothed 11-year moving average (red bold line); Blue filled triangles indicate a forest fire

in nearby area reconstructed from tree-ring fire scars in Mengkeshan (higher) and Pangu (lower),

northern Daxing'an Mountains.

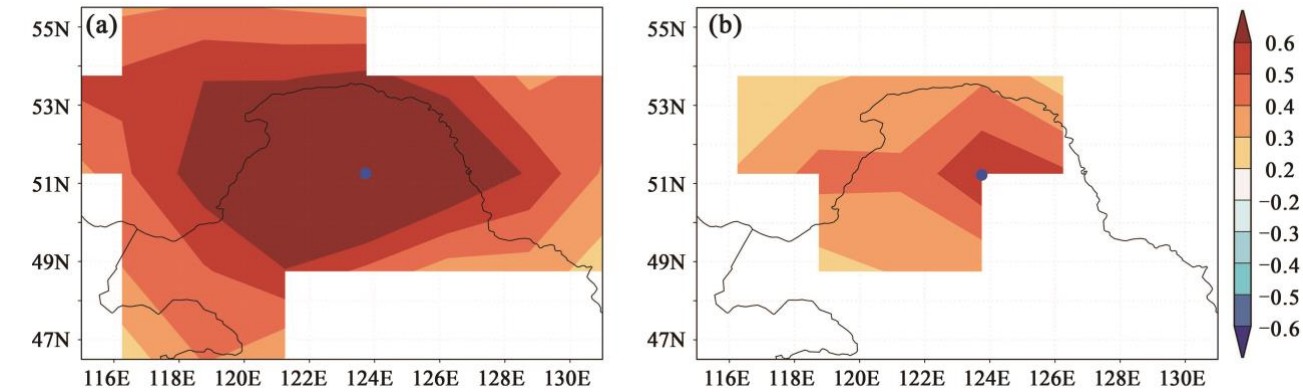


**Fig. 5** Spatial correlation fields between (a) the instrumental and (b) reconstructed annual Dai-PDSI for
the Daxing'an Mountains and the regional Dai-PDSI during the period 1911-2010
(http://climexp.knmi.nl/). The blue circle is the reconstructed PDSI grid.

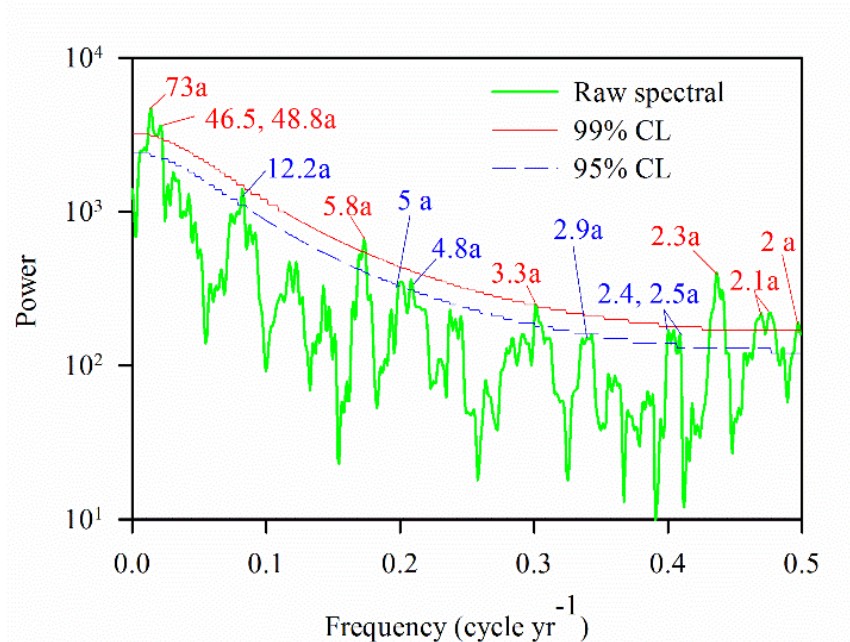


**Fig. 6** Multi-taper method power spectrum of the reconstructed PDSI during the period 1751-2010. The
95% and 99% confidence level relative to red noise are shown and the numbers refer to the significant
period in years.

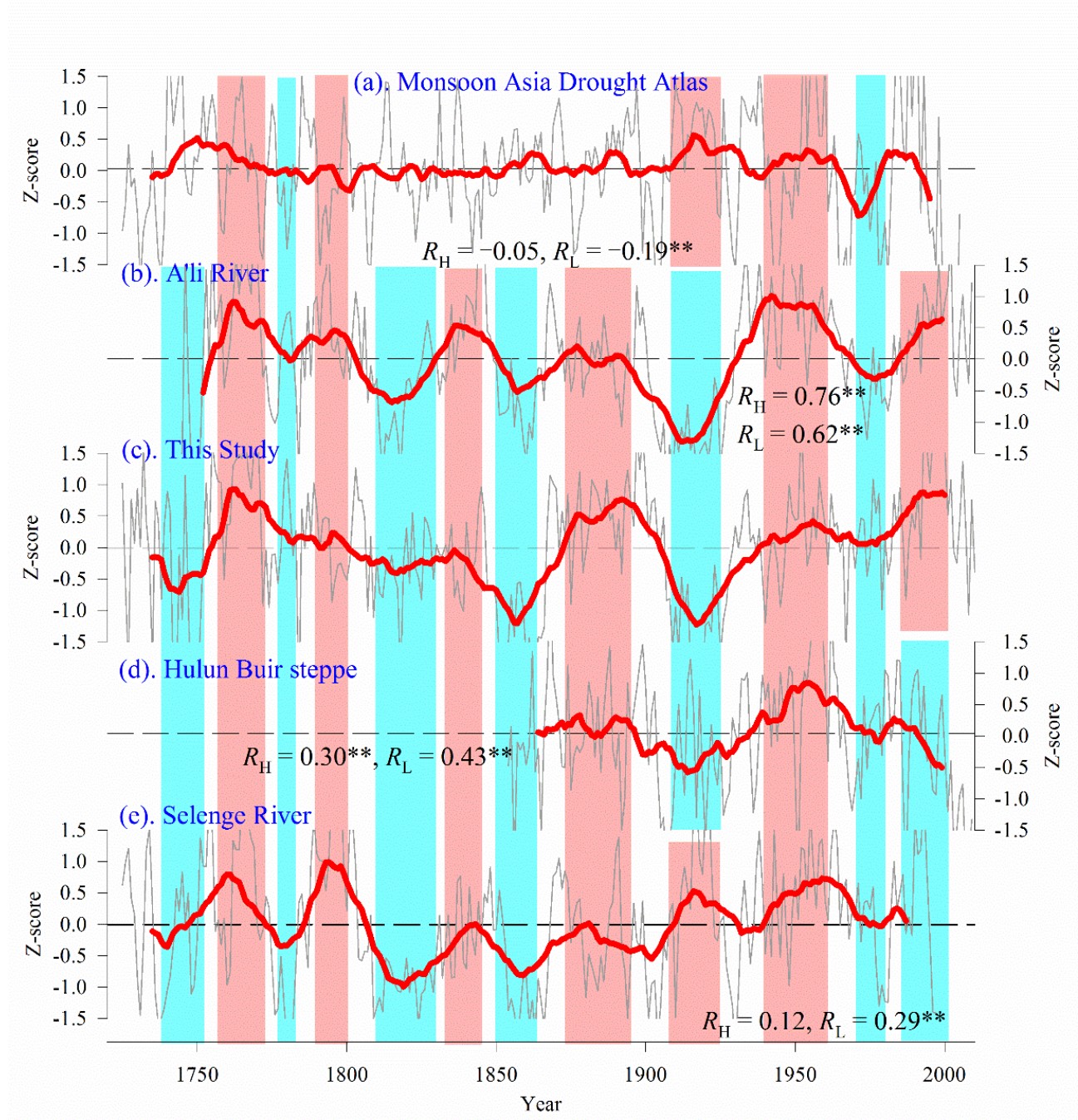


Fig. 7 Comparisons of (a) the drought reconstruction derived from the Monsoon Asia Drought Atlas

(MADA, Cook et al. (2010)), (b) the winter precipitation reconstruction of the A'li River in

northeastern China (AR, Lv and Wang (2014)), (c) the mean annual PDSI reconstruction in the

Daxing'an Mountains (TS, in this study), (d) the April-August SPEI reconstruction of the Hulun Buir

steppe in eastern Mongolian Plateaus (HB, Bao et al. (2015)) and (e) the April-October streamflow

reconstruction of the Selenge River in northeastern Mongolia(SR, Davi et al. (2006)). All above series

were standardized using Z-scores (high frequency) and then smoothed with a 21-year moving average

(low frequency; red bold line). Blue  and red shade areas represent a consistent period of drought and

wetness, respectively. Correlation coefficients between our reconstruction series and other series in low

($R_L$) and high ($R_H$) frequency are shown on the diagram. ** $p < 0.01$

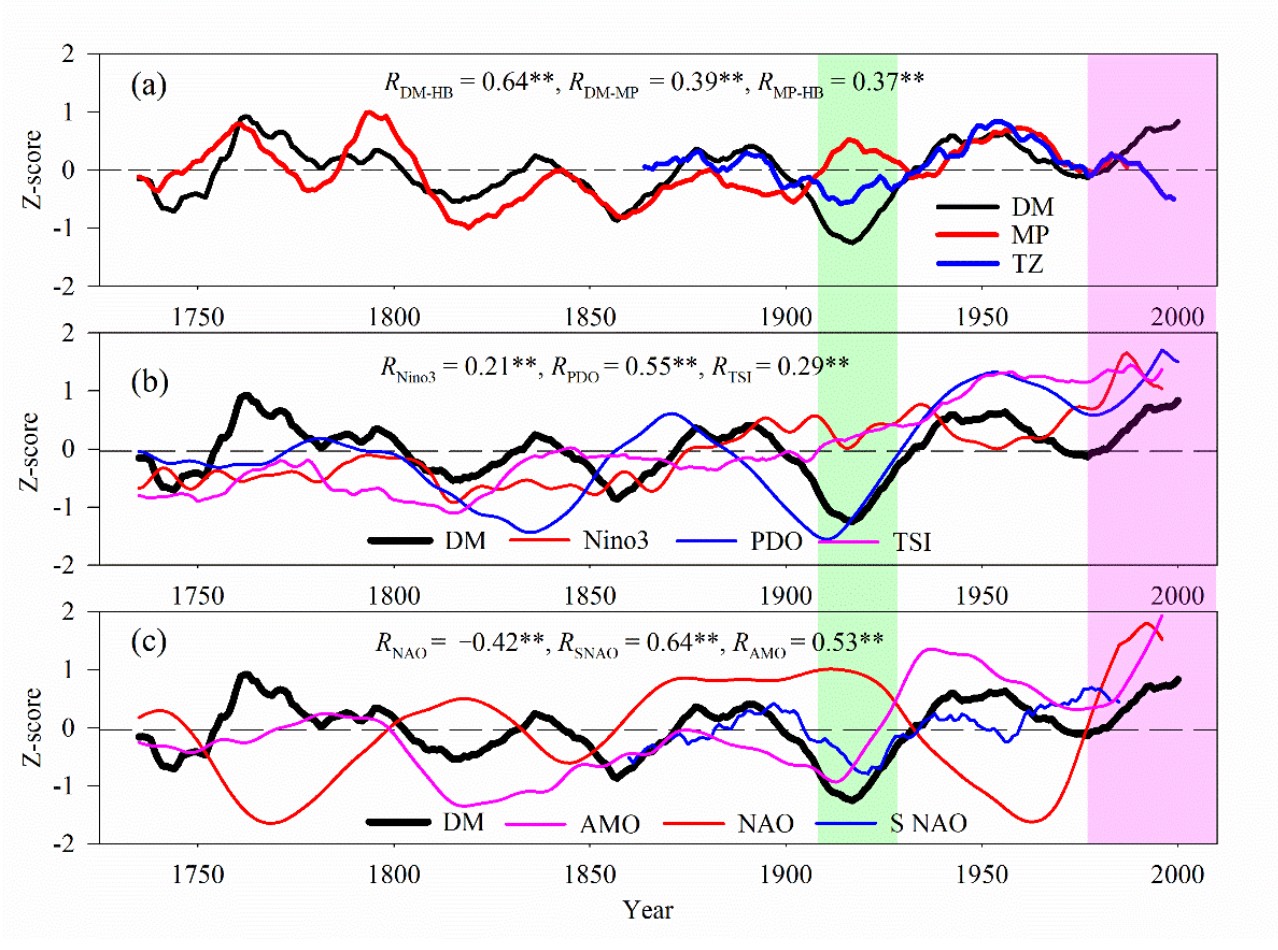

699

Fig. 8 Comparisons of the drought reconstruction and other large-scale climate oscillations. (a) the dry-

wet changes in the Daxing'an Mountains (DM, the average of our reconstruction and the precipitation

reconstruction of the A'li River), the Mongolian Plateaus (MP, the streamflow reconstruction of the

Selenge River) as well as their transition zones (TZ, the SPEI reconstruction of the Hulun Buir steppe);

(b) the drought reconstruction in the Daxing'an Mountains (DM), the Pacific Decadal Oscillation

(PDO) and the Niño 3 index reconstruction from Mann et al. (2009) (Nino3) as well as the Total Solar

Irradiance reconstruction from IPCC AR5 (TSI); (c) the drought reconstruction in the Daxing'an

Mountains (DM), the Atlantic Multidecadal Oscillation reconstruction from Mann et al. (2009) (AMO),

the Multi-decadal Winter North Atlantic Oscillation reconstruction by Trouet et al. (2009)(NAO) and
the summer NAO based on the 20C reanalysis sea-level pressure reconstruction (SNAO). All above
series were standardized using Z-scores and then smoothed with a 21-year moving averaged to highlight
low-frequency drought signals. Significant correlation coefficients (** $p < 0.01$) are listed in the figure.

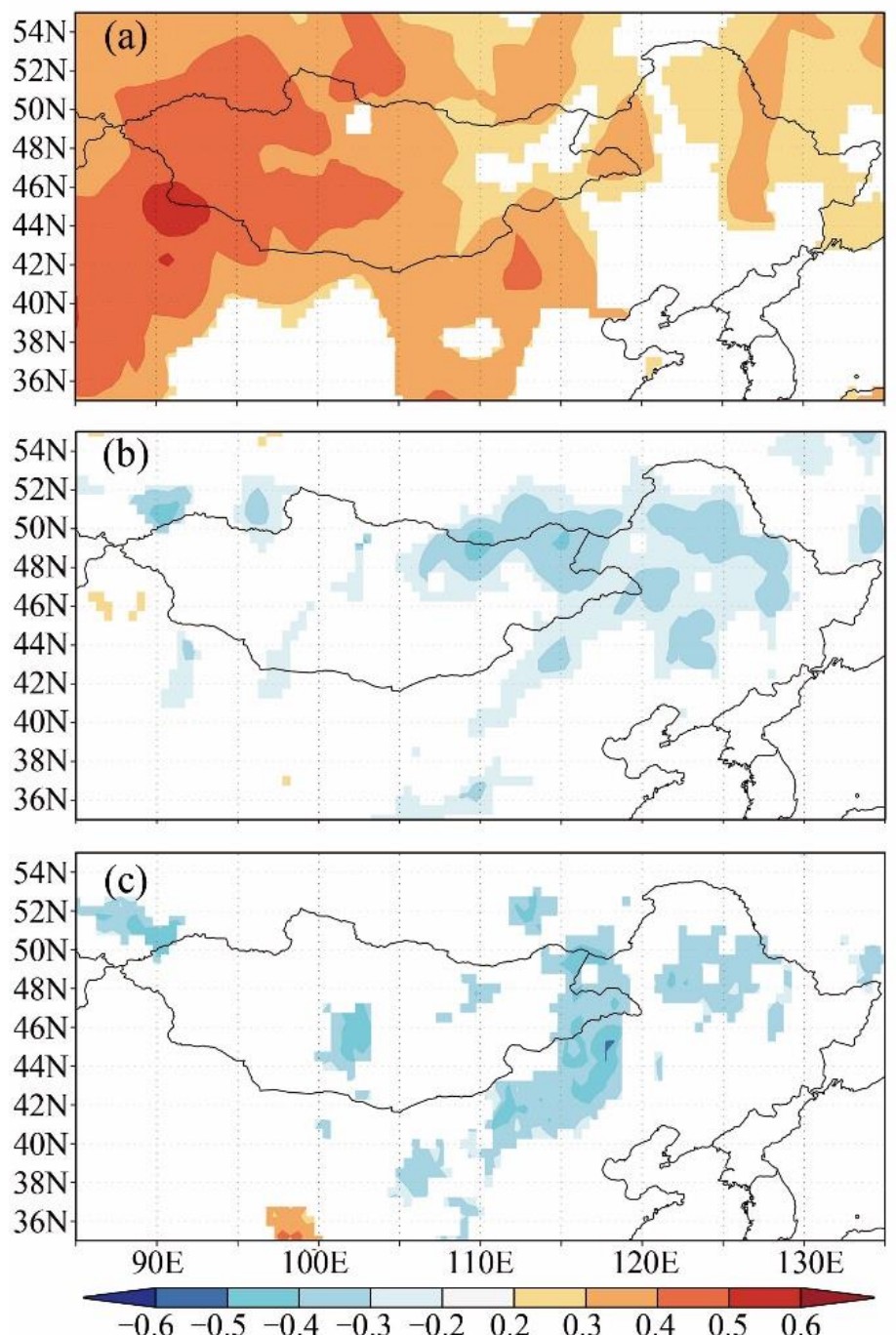


**Fig. 9** Spatial correlations between the annual East Asian monsoon index and the local (a) temperature,

(b) precipitation and (c) scPDSI from AD 1948 to 2010.


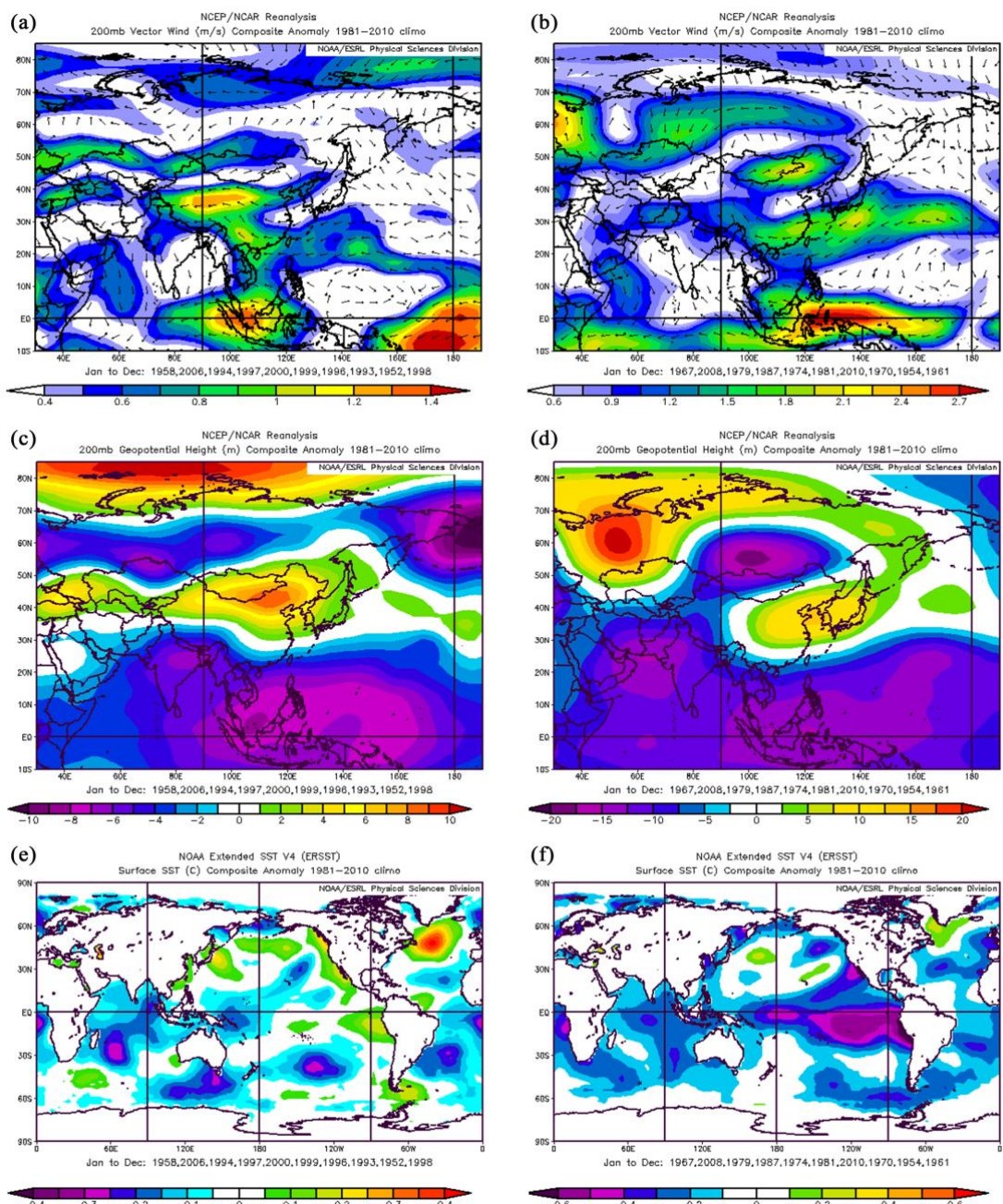

**Fig. 10** Composite anomaly maps of the 200-hPa vector wind and geopotential height, and the SSTs
(from January to December) for the 10 wettest (a, b and e) and 10 driest (c, d and f) years of the Dai-
PDSI reconstruction during the period 1948-2010.