# Peer review of "Response of *Pinus sylvestris* var. *mongolica* to water change and drought history reconstruction in the past 260 years, northeast China"

_Climate of the Past, 2018_

## Referee Comment (RC1) · Anonymous Referee #1 · 30 Apr 2018

This manuscript presented 260-year PDSI reconstruction based on tree-ring record in the central Daxing'an Mountains, NE China. It is a necessary supplement of past climate proxy records in this area, especially for the annual drought reconstruction and its implication for different drought patterns in recent at the Daxing'an Mountains and Mongolian Plateaus (mild drier), NE Asia. Overall this manuscript is well-written, the work seems to be of high quality and is appropriate for Climate of the Past. Therefore, I would recommend this manuscript for publication in this Journal after the following issues are addressed.

1. The manuscript will benefit from a last check by a native speaker. However, readability will improve quite a lot following the careful language check done by reviewer 1. 2. The study shows the drought history of Daxing'an Mountains associated with the Pacific and Atlantic Ocean oscillations, while in the discussion section you linked both PDO and AMO to the Asia Monsoon. Is the PDO or AMO modifing the Asia Monsoon or the Asia Monsoon modifying the PDO? Please check it. 3. In discussion section, the author thinks both the PDO and AMO have the potential to drive or affect the Asian monsoon, which could affect the drought of NE China. Could you give some evidence to prove the Asia Monsoon influence the drought. It's better for you to give some evidence of climate dynamics to prove the mechanism. 4. From the abstruct and conclusion, the readers may feel this tree-ring-based PDSI reconstruction is about the whole region of Daxing'an Mountains, NE China. In fact, it's just a single site PDSI reconstruction. I suggest authors revise it, and specific the study area. For example, just use the central Daxing'an Mountains. 5. In figure 9a, the low frequency MADA series looks not match with its high frequency series. Please check it. 6. It's hard to see the reconstruction point (red) in figure 6, please use different color. 7. Seven tables and twelve figures in your MS, it's too many, some of them could be put in the supplementary materials but in the text. 8. For the mean correlation coefficient between all tree-ring series, use RBAR (in figure 3 and the text) or Rbar (in table 2), please keep it consistent. 9. For the statistic coefficient of correlation, use "R" or "r", please keep it consistent. 10. References in text of the manuscript should be listed in chronological order. 11. Line 48: delete "Therefore". 12. Line 214: replace "A" with "The". 13. Line 261: replace ": The" with ": the". 14. Line 273: delete "transitional". 15. Line 282: replace "by large-scale climate oscillation of the ENSO"with "by ENSO".

---

## Referee Comment (RC2) · Anonymous Referee #2 · 17 May 2018

General comments:

By compositing tree-ring width records from four sites in NE China, the authors have reconstructed a regional PDSI history of past 260 years, and find an increasing PDSI (decreasing drought stress) trend with a warming climate. The historical extreme dry/wet years were identified and discussed. This reconstruction was validated by other drought related reconstructions. The potential impacts of large-scale climate variability, such as AMO, PDO, ENSO etc were tested by correlation analysis.

The long-term regional moisture history is valuable for understanding the response of moisture variability to a warming climate. However, high quality PDSI reconstruction

is still lacking in NE China. The PDSI reconstruction present in this study is based on sufficient tree-ring with data which are produced by standard dendrochronology procedure. The reconstruction results could provide an important insight into the driving mechanisms of PDSI variability of past centuries.

The language of this manuscript needs to be largely improved by native speakers or by professional editing service. I suggest this MS being accepted by CP if the coauthors can address all reviews' concerns and taking into account of comments in CPD (if they are reasonable)

Specific comments:

Lines 9-10: "our reconstruction is accurate and representative, and recorded the same dry years/periods" Does it mean your reconstruction is more accurate and representative than previous ones? If yes, this statement is supported by a more coherence of your reconstruction (than precious ones) with historical and documents and fire history? Line 27 preserved should be replaced by recorded? Line 33 producing should be leading to? Line 40-41 should be improved as: 81 million people and more than 720,000 farmland hectares were suffered from water shortage Line 42-45 River can not on fire, I think you are referring to the Daxing'anling forest fire in May 1987 of Heilongjiang Province, please improve this part correspondingly Line 46-48 This part could be improved as: In order to better character current and project future drought conditions, an improved understanding of past drought variabilities and potential forcing mechanisms is required. However, the short meteorological records of Daxing'an Mountains since the 1950s has limited the understanding of drought history at long-time spectrum. Line 48 remove Therefore Line 48 provide should be serve as Line 52 could improve as: monsoon Asia using 327 tree-ring width chronologies Line 53 could improve as: some disagreements between the MADA results and tree-ring-based local drought reconstructions or instrumental drought data, especially in eastern Asia, which might due to an insufficient tree-ring network used by MADA (Li et al. 2015; Liu et al. 2016). Line 66 farther should be further? Line 67 should be clarified Line 70

high-latitude forested portion should be high-latitude forests Line 73 remove extreme Line 111 should be improved as: Pearson correlation analysis was conducted to esita- mate climate–tree growth relationships Line 112 should be improved as: The gridded climate dataset is much longer and has higher homogeneity and coherency than sta- tion data. Line 115-116 remove ", a most commonly used drought index," Line 146 you should specify here which "large-scale climate" indexes are tested Line 162 is it ok to substitute "PDSI data among the annual, seasonal or individual month scales" with all seasonal PDSI compositions? Line 166-167 is it ok to replace "The regression model between the tree-ring indices (predictors) and annual PDSI (predicted) for the calibration period was as follows" with The linear model for PDSI reconstruction is? Line 171 please replace "actual" and "estimated" with instrumental and reconstructed throughout this manuscript Line 173-174 is figure 6a a correlation between one PDSI index with another PDSI index? I think figure 6a and 6b could be replace a spatial correlation map between PDSI reconstruction and dai-PDSI. Line175 please specify which two calibration periods. Line175-178 please move this section after "is the tree- ring index at year t." of line 169, and add a sentence at the end of this paragraph, such as: suggesting this linear model is robust for PDSI reconstruction. Line 170-174 should be another paragraph after "suggesting this linear model is robust for PDSI reconstruction" Line 179 please replace "Drought-wet variations" with historical PDSI variability Line 188 replace greatest with greater Line 191 and 193 Table 5 only show the individual dry/wet years, consecutive dry/wet periods was absent in Table 5, is that right? Line 196 replace "the dry and wet variations" with historical PDSI variability Line 202-203 replace "main climate limitation for its radial growth" main climate factor limit- ing its radial growth Line 218-219 this section is not clear so far. Is it possible that the positive correlation of tree growth and winter temperature could arise from less frost damage if the winter temperature is higher? Is Scots pine in your study an evergreen tree species? If yes, the positive correlation of tree growth and winter temperature could also because higher winter photosynthetic rates and more photosynthetic prod- ucts stored if temperature is high in winter, these photosynthetic products will be used

for tree growth in summer of next year (storage effect). The positive correlation with spring temperature could due to earlier and larger snow melting which supplies the spring soil water, and eventually stimulated tree growth? Anyway this section should be improved accordingly. Line 225-226 is the "local historical record" and "historical documents" have been specified in the Data and Methods part? Since I can't find them before section 4.2. Line 234 Are PDSI reconstructions in Mengkeshan and Pangu from your data, or previous studies of other people? If they are from previous studies outside your sampling region, is it possible to do the same SEA analysis with your own data of this study? Line 243-253 I agree with you that Cook's MADA reconstruction is inaccurate and sometime useless in regions with no or a few tree-ring data, such as your study region.
* * *

---

## Editor Comment (EC1) · N. Combourieu Nebout (Editor) · 28 May 2018

Dear authors,

We have now received the response of the two reviewers. Both of them address several scientific comments on your manuscript and underline the problem of english language.

You have now to post your replies to all the comments on the discussion forum in which you have to explain how you want to modify your manuscript. Please also prepare your responses in track mode change.

Waiting after your response to the reviewers

[Figure]

With my best regards

Nathalie Combourieu-Nebout

---

## Author Comment (AC1) · 3 Jun 2018

Comments and Response

Anonymous Referee #2

General comments:

By compositing tree-ring width records from four sites in NE China, the authors have reconstructed a regional PDSI history of past 260 years, and find an increasing PDSI (decreasing drought stress) trend with a warming climate. The historical extreme dry/wet years were identified and discussed. This reconstruction was validated by

other drought related reconstructions. The potential impacts of large-scale climate variability, such as AMO, PDO, ENSO etc. were tested by correlation analysis. The long-term regional moisture history is valuable for understanding the response of moisture variability to a warming climate. However, high quality PDSI reconstruction is still lacking in NE China. The PDSI reconstruction present in this study is based on sufficient tree-ring with data which are produced by standard dendrochronology procedure. The reconstruction results could provide an important insight into the driving mechanisms of PDSI variability of past centuries. The language of this manuscript needs to be largely improved by native speakers or by professional editing service. I suggest this MS being accepted by CP if the coauthors can address all reviews' concerns and taking into account of comments in CPD (if they are reasonable)

Response: we agree. This MS will be revised by a native English speaker. We will go through the MS and improve the language.

Specific comments:

Lines 9-10: "our reconstruction is accurate and representative, and recorded the same dry years/periods" Does it mean your reconstruction is more accurate and representative than previous ones? If yes, this statement is supported by a more coherence of your reconstruction (than precious ones) with historical and documents and fire history?

Response: we accepted. We have revised it as: Our reconstruction is coherence with local historical documents and other nearby hydroclimate reconstructions.

Line 27 preserved should be replaced by recorded?

Response: thanks, we have done.

Line 33 producing should be leading to?

Response: thanks, we have done.

Line 40-41 should be improved as: 81 million people and more than 720,000 farmland hectares were suffered from water shortage

Response: thanks, we have done.

Line 42-45 River can not on fire, I think you are referring to the Daxing'anling forest fire in May 1987 of Heilongjiang Province, please improve this part correspondingly.

Response: we accepted. We have revised it as: In addition, drought is in favour the occurrence of large wildfires, and the drought of Daxing'an Mountains especially in spring and summer is often accompanied by high risk of forest wildfire disasters and (Sun 2007). For example, the forest fire in May 1987 killed over 200 people and burned ~73,000 km2 (Sun 2007; Yao et al. 2017).

Line 46-48 This part could be improved as: In order to better character current and project future drought conditions, an improved understanding of past drought variabilities and potential forcing mechanisms is required. However, the short meteorological records of Daxing'an Mountains since the 1950s has limited the understanding of drought history at long- time spectrum.

Response: we accepted. We have done.

Line 48 remove Therefore

Response: we accepted. We have done.

Line 48 provide should be serve as

Response: we accepted. We have done.

Line 52 could improve as: monsoon Asia using 327 tree-ring width chronologies

Response: we accepted. We have done.

Line 53 could improve as: some disagreements between the MADA results and tree-ring-based local drought reconstructions or instrumental drought data, especially in

eastern Asia, which might due to an insufficient tree-ring network used by MADA (Li et al. 2015; Liu et al. 2016).

Response: we accepted. We have done.

Line 66 farther should be further?

Response: we accepted. We have done.

Line 67 should be clarified

Response: we accepted. We have done.

Line 70 high-latitude forested portion should be high-latitude forests

Response: we accepted. We have done.

Line 73 remove extreme

Response: we accepted. We have done.

Line 111 should be improved as: Pearson correlation analysis was conducted to estimate climate–tree growth relationships

Response: we accepted. We have done.

Line 112 should be improved as: The gridded climate dataset is much longer and has higher homogeneity and coherency than station data.

Response: we accepted. We have done.

Line 115-116 remove ", a most commonly used drought index,"

Response: we accepted. We have done.

Line 146 you should specify here which "large-scale climate" indexes are tested

Response: we accepted. We have done.

Line 162 is it ok to substitute "PDSI data among the annual, seasonal or individual month scales" with all seasonal PDSI compositions?

Response: we accepted. We have done.

Line 166-167 is it ok to replace "The regression model between the tree-ring indices (predictors) and annual PDSI (predicted) for the calibration period was as follows" with the linear model for PDSI reconstruction is?

Response: we accepted. We have done.

Line 171 please replace "actual" and "estimated" with instrumental and reconstructed throughout this manuscript

Response: we accepted. We have done.

Line 173-174 is figure 6a a correlation between one PDSI index with another PDSI index? I think figure 6a and 6b could be replace a spatial correlation map between PDSI reconstruction and dai-PDSI.

Response: we accepted. We have changed figure 6.

Line175 please specify which two calibration periods.

Response: we accepted. We have done.

Line175-178 please move this section after "is the tree- ring index at year t." of line 169, and add a sentence at the end of this paragraph, such as: suggesting this linear model is robust for PDSI reconstruction.

Response: we accepted. We have done.

Line 170-174 should be another paragraph after "suggesting this linear model is robust for PDSI reconstruction"

Response: we accepted. We have done.

[Figure]

Line 179 please replace "Drought-wet variations" with historical PDSI variability

Response: we accepted. We have done.

Line 188 replace greatest with greater

Response: we accepted. We have done.

Line 191 and 193 Table 5 only show the individual dry/wet years, consecutive dry/wet periods were absented in Table 5, is that right?

Response: we accepted. In fact, the consecutive dry/wet periods were shown in Table 6. We have revised it in the text.

Line 196 replace "the dry and wet variations" with "historical PDSI variability".

Response: we accepted. We have done.

Line 202-203 replace "main climate limitation for its radial growth" with "main climate factor limiting its radial growth".

Response: we accepted. We have done.

Line 218-219 this section is not clear so far. Is it possible that the positive correlation of tree growth and winter temperature could arise from less frost damage if the winter temperature is higher? Is Scots pine in your study an evergreen tree species? If yes, the positive correlation of tree growth and winter temperature could also because higher winter photosynthetic rates and more photosynthetic products stored if temperature is high in winter, these photosynthetic products will be used for tree growth in summer of next year (storage effect). The positive correlation with spring temperature could due to earlier and larger snow melting which supplies the spring soil water, and eventually stimulated tree growth? Anyway, this section should be improved accordingly.

Response: we accepted. we have revised this section according you comment:

On the contrary, a significant positive response of radial growth to non-growing season

temperature was found. It is possible that higher winter photosynthetic rates and more photosynthetic products stored in warm winter could be used for tree growth in summer of next year (storage effect), and higher winter temperature could also arise from less frost damage; the positive correlation with spring temperature could due to earlier and larger snow melting which supplies the spring soil water, and eventually stimulated tree growth (Hollesen et al. 2015; Zhu et al. 2017).

Line 225-226 is the "local historical record" and "historical documents" have been specified in the Data and Methods part? Since I can't find them before section 4.2. Response: we accepted. We added the description of those data in the Methods section: Local historical drought data recorded in book "Meteorological disasters dictionary of China" (Shen 2008; Sun 2007) were used to verify our PDSI reconstruction.

Line 234 Are PDSI reconstructions in Mengkeshan and Pangu from your data, or previous studies of other people? If they are from previous studies out- side your sampling region, is it possible to do the same SEA analysis with your own data of this study?

Response: In fact, the data in Mengkeshan and Pangu which were used for SEA analysis are the forest fire event data. The fire event data were reconstructed by fire scars (Yao et al., 2017). Forest fires usually occur in dry years, so the occurrence of forest fires can reflect drought events from the side. The SEA analysis between forest fire history and reconstructed drought variables could further validate the accuracy of our reconstruction.

Line 243-253 I agree with you that Cook's MADA reconstruction is inaccurate and sometime useless in regions with no or a few tree-ring data, such as your study region.

Response: thanks. Although, the divergence of between the MADA and individual tree-ring-based drought reconstruction has been found in some studies (Li et al., 2015; Liu et al., 2016; and this study), more evidence is still urgently needed.

---

## Author Comment (AC2) · 3 Jun 2018

Comments and Response:

Anonymous Referee #1

This manuscript presented 260-year PDSI reconstruction based on tree-ring record in the central Daxing'an Mountains, NE China. It is a necessary supplement of past climate proxy records in this area, especially for the annual drought reconstruction and its implication for different drought patterns in recent at the Daxing'an Mountains and Mongolian Plateaus (mild drier), NE Asia. Overall this manuscript is well-written, the

work seems to be of high quality and is appropriate for Climate of the Past. Therefore, I would recommend this manuscript for publication in this Journal after the following issues are addressed.

C1. The manuscript will benefit from a last check by a native speaker. However, readability will improve quite a lot following the careful language check done by reviewer 2.

Response: we agree. The MS will be revised by the native English speaker. We will go through the MS and make sure the language suitable for publish.

C2. The study shows the drought history of Daxing'an Mountains associated with the Pacific and Atlantic Ocean oscillations, while in the discussion section you linked both PDO and AMO to the Asia Monsoon. Is the PDO or AMO modifying the Asia Monsoon or the Asia Monsoon modifying the PDO? Please check it.

Response: we accepted. We have double check the describe. We make sure the PDO or AMO could modify the Asia Monsoon (Ma et al., 2007; Cook et al., 2010; Li et al., 2015; Linderholm et al., 2011; Sun et al., 2008; Chen et al., 2015; Bao et al., 2015).

C3. In discussion section, the author thinks both the PDO and AMO have potential to drive or affect the Asian monsoon, which could affect the drought of NE China. Could you give some evidence to prove the Asia Monsoon influence the drought. It's better for you to give some evidence of climate dynamics to prove the mechanism.

Response: we accepted. We have added the composite anomaly maps of the 200-hPa vector wind and geopotential height, and the SSTs (from January to December) for the 10 wettest and 10 driest years for the Dai-PDSI reconstruction during the period 1948–2010. Some new explanations (following) of climate dynamics were added in discussion section.

Previous studies have found the drought variation of northeast Asia may be teleconnected with the activity of the Asian monsoon (Cook et al., 2010;Li et al., 2015;Linderholm et al., 2011;Sun et al., 2008;Chen et al., 2015;Bao et al., 2015). During the wet years, the strengthened southerlies and easterlies entered China inland associated with a positive pattern over northeast Asia and some negative height-anomaly centers in west Russia and south Asia as well as the Indian and north Pacific oceans, which strengthened the westerly circulation (Fig. 10a, c). For the dry-year composite, however, strengthened southerlies and south-westerlies entered northeast China associated with a positive pattern over east Asia and western Russia, and some negative height-anomaly centers in southern Russia and south Asia as well as the Indian and south Pacific oceans (Fig. 10a, c).

The composite of 200-hPa geopotential height over the Central-North Daxing'an Mountains during the wettest 10 years (positive anomaly) is the reverse of the driest 10 years (negative anomaly) (Fig. 1c, d). Positive and negative SST anomalies were found in the western and northern Pacific Ocean during the wettest and driest years (Fig. 1e, f). During the wet years, relatively abundant moisture is brought from the Pacific Ocean by the strong east Asian monsoon (southeasterly moisture flux), and traveled further northward to Mongolian Plateaus, and then joined with a strong Westerly circulation that causing increased moisture over the Daxing'an Mountains (Fig. 1a). This negative anomaly combined with positive SST in the western and northern Pacific Ocean leads to an enhanced dry jet (south-westerlies) across/toward the Daxing'an Mountains (Fig. 1b, c, e). Similar finding that drought variations over northeast Asia are strongly linked with the Asian monsoon and SSTs in the Pacific and Atlantic oceans were also found by (Cook et al., 2010;Li et al., 2015;Linderholm et al., 2011;Chen et al., 2015;Bao et al., 2015) . Besides, the abnormally low potential evaporation pattern in Daxing'an Mountains during the wettest years supports such a connection or pattern (Fig. 2). Therefore, dry-wet conditions in Daxing'an Mountains are linked with SSTs in the Pacific and Atlantic oceans and Asian monsoon intensity.

C4. From the abstract and conclusion, the readers may feel this tree-ring-based PDSI reconstruction is about the whole region of Daxing'an Mountains, NE China. In fact,

it's just a single site PDSI reconstruction. I suggest authors revise it, and specific the study area. For example, just use the central Daxing'an Mountains.

Response: we accepted. We have revised it.

C5. In figure 9a, the low frequency MADA series looks not match with its high frequency series. Please check it.

Response: we have double check the data. We make sure that it's right.

C6. It's hard to see the reconstruction point (red) in figure 6, please use different color.

Response: we accepted. We have changed the figure.

C7. Seven tables and twelve figures in your MS, it's too many, some of them could be put in the supplementary materials but in the text.

Response: we accepted. We have moved some figures (figure 3, 8 and 11) and tables (Table 2) to the supplementary materials. Now only six tables and ten figures is left.

C8. For the mean correlation coefficient between all tree-ring series, use RBAR (in figure 3 and the text) or Rbar (in table 2), please keep it consistent.

Response: we accepted. We have done.

C9. For the statistic coefficient of correlation, use "R" or "r", please keep it consistent.

Response: we accepted. We have done.

C10. References in text of the manuscript should be listed in chronological order.

Response: we accepted. We have done.

C11. Line 48: delete "Therefore".

Response: we accepted. We have done.

C12. Line 214: replace "A" with "The".

Response: we accepted. We have done.

C13. Line 261: replace ": The" with ": the".

Response: we accepted. We have done.

C14. Line 273: delete "transitional".

Response: we accepted. We have delete it.

C15. Line 282: replace "by large-scale climate oscillation of the ENSO"with "by ENSO".

Response: we accepted. We have done.

[Figure]

**Fig. 1.** Composite of the 200-hPa vector wind and Geopotential height, and SSTs (from Jan to Dec) for the 10 wettest (left) and 10 driest (right) years for the Dai-PDSI reconstruction during the 1948-2010.

[Figure]

**Fig. 2.** Composite anomaly maps of the surface potential evaporation (W/mˆ2) (from Jan to Dec) for the 10 wettest (a) and 10 driest (b) years for the Dai-PDSI reconstruction during the period 1948–2010.

---

## Author Comment (AC3) · 3 Jun 2018

Thank you for reminding us. We have posted the replies to all the comments on the discussion forum.

---

## Referee Report (RR1)

Dear authors:

Thank you so much for the point by point replies and corresponding revisions. All my concerns have been properly clarified, and the language quality has been largely improved. I am satisfied with the current manuscript and suggesting this version could be published in CP as it is.

---

## Author Response (AR2)

**Dear Dr. Nathalie and reviewers,**

Thank you for giving us a chance to revise our manuscript (cp-2018-31). We greatly appreciate the two anonymous reviewers and you for your valuable and helpful comments, which greatly improved the quality of our manuscript. All comments were fully accepted, and we have revised our manuscript according them. The point by point response to the reviewers' comments are in the following sections. The reviewers' comments are listed in blue, and our responses are in black. Thanks again for all your help in processing our manuscript.

Best wishes,

Xiaochun Wang

On behalf of the authors.

Corresponding author: Xiaochun Wang at Center for Ecological Research, School of Forestry, Northeast Forestry University, Harbin 150040, China,

Phone: +86 451 82190615, E-mail address: wangx@nefu.edu.cn.

**Response to anonymous Referees**

\*\*\*\*\*\*\*\*\*\*\*\*\*\*\*\*\*\*\*\*\*\*\*\*\*\*\*\*\*\*\*\*\*\*\*\*\*\*\*\*\*\*\*\*\*\*\*\*\*\*\*\*\*\*\*\*\*\*\*\*\*\*\*\*\*\*\*\*\*\*\*\*\*\*\*\*\*\*\*\*\*\*\*

**Anonymous Referee #1**

This manuscript presented 260-year PDSI reconstruction based on tree-ring record in the central Daxing'an Mountains, NE China. It is a necessary supplement of past climate proxy records in this area, especially for the annual drought reconstruction and its implication for different drought patterns in recent at the Daxing'an Mountains and Mongolian Plateaus (mild drier), NE Asia. Overall this manuscript is well-written, the work seems to be of high quality and is appropriate for Climate of the Past. Therefore, I would recommend this manuscript for publication in this Journal after the following issues are addressed.

C1. The manuscript will benefit from a last check by a native speaker. However, readability will improve quite a lot following the careful language check done by reviewer 2.

**Response: Fully accepted**. The MS will be revised by the native English speaker. We will go through the MS and make sure the language suitable for publish.

C2. The study shows the drought history of Daxing'an Mountains associated with the Pacific and Atlantic Ocean oscillations, while in the discussion section you linked both PDO and AMO to the Asia Monsoon. Is the PDO or AMO modifying the Asia Monsoon or the Asia Monsoon modifying the PDO? Please check it.

**Response: Fully accepted**. We have double check the describe. We make sure the PDO or AMO could modify the Asia Monsoon (Ma et al., 2007; Cook et al., 2010; Li et al., 2015; Linderholm et al., 2011; Sun et al., 2008; Chen et al., 2015; Bao et al., 2015).

C3. In discussion section, the author thinks both the PDO and AMO have potential to drive or affect the Asian monsoon, which could affect the drought of NE China. Could you give some evidence to prove the Asia Monsoon influence the drought. It's better for you to give some evidence of climate dynamics to prove the mechanism.

**Response: Fully accepted**. We have added the composite anomaly maps of the 200-hPa vector wind and geopotential height, and the SSTs (from January to December) for the 10 wettest and 10 driest years for the Dai-PDSI reconstruction during the period 1948–2010. Some new explanations (following) of climate dynamics were added in discussion section.

Previous studies have found that drought variation in northeast Asia may be associated with Asian monsoon activity (Bao et al., 2015; Chen et al., 2015; Cook et al., 2010; Li et al., 2015; Linderholm et al., 2011; Sun et al., 2008). In wet years, the strengthened southerlies and easterlies entered inland China associated with a positive pattern over northeast Asia and some negative height-anomaly centers in west Russia and south Asia as well as the Indian and north Pacific oceans, which strengthened the westerly circulation (Fig. 10a, c). In dry years, however, strengthened southerlies and south-westerlies entered northeast China associated with a positive pattern over east

Asia and western Russia, and some negative height-anomaly centers in southern Russia and south Asia as well as the Indian and south Pacific oceans (Fig. 10a, c).

The composite of 200-hPa geopotential height of the most humid 10 years (positive anomaly) in the central-north Daxing'an Mountains is opposite to that of the most arid 10 years (negative anomaly) (Fig. 10c, d). Positive and negative SST anomalies were also found in the western and northern Pacific Ocean during the wettest and driest years (Fig. 10e, f). In the wet years, abundant moisture is transported from the Pacific Ocean through Mongolian Plateau to the Daxing'an Mountains *via* the strong east Asian monsoon's southeasterly moisture flux joined with a strong Westerly circulation (Fig. 10a). This negative anomaly combined with positive SST in the western and northern Pacific Ocean lead to an enhanced dry jet (south-westerlies) across/toward the Daxing'an Mountains (Fig. 10b, c, e). Several studies have reported that the dry and wet variations in northeast Asia are strongly linked with the Asian monsoon and SSTs in the Pacific and Atlantic oceans (Bao et al., 2015; Chen et al., 2015; Cook et al., 2010; Li et al., 2015; Linderholm et al., 2011). In addition, the potential evaporation pattern in the Daxing'an Mountains is extremely low in the wettest years, and it also supports the above remote-connection assumptions (Fig. S4).

[Figure]

**Fig. 10** Composite anomaly maps of the 200-hPa vector wind and geopotential height, and the SSTs (from January to December) for the 10 wettest (a, b and e) and 10 driest (c, d and f) years of the Dai-PDSI reconstruction during the period 1948-2010.

[Figure]

**Fig. S4** Composite anomaly maps of the surface potential evaporation (W/m^2) (from January to December) for the 10 wettest (a) and 10 driest (b) years for the Dai-PDSI reconstruction during the period 1948–2010.

C4. From the abstract and conclusion, the readers may feel this tree-ring-based PDSI reconstruction is about the whole region of Daxing'an Mountains, NE China. In fact, it's just a single site PDSI reconstruction. I suggest authors revise it, and specific the study area. For example, just use the central Daxing'an Mountains.
**Response: Fully accepted**. We have revised it.

C5. In figure 9a, the low frequency MADA series looks not match with its high frequency series. Please check it.
**Response:** we have double check the data. We make sure that it's right.

C6. It's hard to see the reconstruction point (red) in figure 6, please use different color.
**Response: Fully accepted**. We have changed the figure. The new figure is following:

[Figure]

**Fig. 5** Spatial correlation fields between (a) the instrumental and (b) reconstructed annual Dai-PDSI for the Daxing'an Mountains and the regional Dai-PDSI during the period 1911-2010 (http://climexp.knmi.nl/). The blue circle is the reconstructed PDSI grid.

C7. Seven tables and twelve figures in your MS, it's too many, some of them could be put in the supplementary materials but in the text.

**Response: Fully accepted.** We have moved some figures (figure 3, 8 and 11) and tables (Table 2) to the supplementary materials. Now only six tables and ten figures is left.

C8. For the mean correlation coefficient between all tree-ring series, use RBAR (in figure 3 and the text) or Rbar (in table 2), please keep it consistent.
**Response: Fully accepted.** Done.

C9. For the statistic coefficient of correlation, use "*R*" or "*r*", please keep it consistent.
**Response: Fully accepted.** Done.

C10. References in text of the manuscript should be listed in chronological order.
**Response: Fully accepted.** Done.

C11. Line 48: delete "Therefore".
**Response: Fully accepted.** Done.

C12. Line 214: replace "A" with "The".

**Response: Fully accepted.** Done.

C13. Line 261: replace ": The" with ": the".

**Response: Fully accepted.** Done.

C14. Line 273: delete "transitional".

**Response: Fully accepted.** Done.

C15. Line 282: replace "by large-scale climate oscillation of the ENSO"with "by ENSO".

**Response: Fully accepted.** Done.

**Anonymous Referee #2**

General comments:

By compositing tree-ring width records from four sites in NE China, the authors have reconstructed a regional PDSI history of past 260 years, and find an increasing PDSI (decreasing drought stress) trend with a warming climate. The historical extreme dry/wet years were identified and discussed. This reconstruction was validated by other drought related reconstructions. The potential impacts of large-scale climate variability, such as AMO, PDO, ENSO etc. were tested by correlation analysis.

The long-term regional moisture history is valuable for understanding the response of moisture variability to a warming climate. However, high quality PDSI reconstruction is still lacking in NE China. The PDSI reconstruction present in this study is based on sufficient tree-ring with data which are produced by standard dendrochronology procedure. The reconstruction results could provide an important insight into the driving mechanisms of PDSI variability of past centuries.

The language of this manuscript needs to be largely improved by native speakers or by professional editing service. I suggest this MS being accepted by CP if the coauthors can address all reviews' concerns and taking into account of comments in CPD (if they are reasonable)

**Response: Fully accepted**. The third coauthor (David Cooper) of this MS is a native English speaker. He has gone through the MS and improved the language.

Specific comments:

Lines 9-10: "our reconstruction is accurate and representative, and recorded the same dry years/periods" Does it mean your reconstruction is more accurate and representative than previous ones? If yes, this statement is supported by a more coherence of your reconstruction (than precious ones) with historical and documents and fire history?

**Response: Fully accepted.** We have revised it as: Our reconstruction is coherence with local historical documents and other nearby hydroclimate reconstructions.

Line 27 preserved should be replaced by recorded?

**Response: Fully accepted.** Done.

Line 33 producing should be leading to?

**Response: Fully accepted.** Done.

Line 40-41 should be improved as: 81 million people and more than 720,000 farmland hectares were suffered from water shortage

**Response: Fully accepted.** Done.

Line 42-45 River cannot on fire, I think you are referring to the Daxing'anling forest fire in May 1987 of Heilongjiang Province, please improve this part correspondingly.

**Response:** we accepted. We have revised it as: In addition, drought is in favor the occurrence of large wildfires, and the drought of Daxing'an Mountains especially in spring and summer is often accompanied by high risk of forest wildfire disasters and (Sun 2007). For example, the forest fire in May 1987 killed over 200 people and burned ~73,000 km2 (Sun 2007; Yao et al. 2017).

Line 46-48 This part could be improved as: In order to better character current and project future drought conditions, an improved understanding of past drought variabilities and potential forcing mechanisms is required. However, the short

meteorological records of Daxing'an Mountains since the 1950s has limited the understanding of drought history at long- time spectrum.

**Response: Fully accepted.** Done.

Line 48 remove Therefore

**Response: Fully accepted.** Done.

Line 48 provide should be serve as

**Response: Fully accepted.** Done.

Line 52 could improve as: monsoon Asia using 327 tree-ring width chronologies

**Response: Fully accepted.** Done.

Line 53 could improve as: some disagreements between the MADA results and tree-ring-based local drought reconstructions or instrumental drought data, especially in eastern Asia, which might due to an insufficient tree-ring network used by MADA (Li et al. 2015; Liu et al. 2016).

**Response: Fully accepted.** Done.

Line 66 farther should be further?

**Response: Fully accepted.** Done.

Line 67 should be clarified

**Response: Fully accepted.** Done.

Line 70 high-latitude forested portion should be high-latitude forests

**Response: Fully accepted.** Done.

Line 73 remove extreme

**Response: Fully accepted.** Done.

Line 111 should be improved as: Pearson correlation analysis was conducted to estimate climate–tree growth relationships

**Response: Fully accepted.** Done.

Line 112 should be improved as: The gridded climate dataset is much longer and has higher homogeneity and coherency than station data.
**Response: Fully accepted.** Done.

Line 115-116 remove ", a most commonly used drought index,"
**Response: Fully accepted.** Done.

Line 146 you should specify here which "large-scale climate" indexes are tested
**Response: Fully accepted.** Done.

Line 162 is it ok to substitute "PDSI data among the annual, seasonal or individual month scales" with all seasonal PDSI compositions?
**Response: Fully accepted.** Done.

Line 166-167 is it ok to replace "The regression model between the tree-ring indices (predictors) and annual PDSI (predicted) for the calibration period was as follows" with the linear model for PDSI reconstruction is?
**Response: Fully accepted.** Done.

Line 171 please replace "actual" and "estimated" with instrumental and reconstructed throughout this manuscript
**Response: Fully accepted.** Done.

Line 173-174 is figure 6a a correlation between one PDSI index with another PDSI index? I think figure 6a and 6b could be replace a spatial correlation map between PDSI reconstruction and dai-PDSI.
**Response: Fully accepted.** We have changed figure 6.

[Figure]

**Fig. 5** Spatial correlation fields between (a) the instrumental and (b) reconstructed annual Dai-PDSI for the Daxing'an Mountains and the regional Dai-PDSI during the period 1911-2010 (http://climexp.knmi.nl/). The blue circle is the reconstructed PDSI grid.

Line175 please specify which two calibration periods.
**Response: Fully accepted.** Done.

Line175-178 please move this section after "is the tree- ring index at year t." of line 169, and add a sentence at the end of this paragraph, such as: suggesting this linear model is robust for PDSI reconstruction.
**Response: Fully accepted**. Done.

Line 170-174 should be another paragraph after "suggesting this linear model is robust for PDSI reconstruction"
**Response: Fully accepted**. Done.

Line 179 please replace "Drought-wet variations" with historical PDSI variability
**Response: Fully accepted**. Done.

Line 188 replace greatest with greater
**Response: Fully accepted**. Done.

Line 191 and 193 Table 5 only show the individual dry/wet years, consecutive dry/wet periods were absented in Table 5, is that right?

**Response: Fully accepted**. In fact, the consecutive dry/wet periods were shown in Table 6. We have revised it in the text.

Line 196 replace "the dry and wet variations" with "historical PDSI variability".
**Response: Fully accepted**. Done.

Line 202-203 replace "main climate limitation for its radial growth" with "main climate factor limiting its radial growth".
**Response: Fully accepted**. Done.

Line 218-219 this section is not clear so far. Is it possible that the positive correlation of tree growth and winter temperature could arise from less frost damage if the winter temperature is higher? Is Scots pine in your study an evergreen tree species? If yes, the positive correlation of tree growth and winter temperature could also because higher winter photosynthetic rates and more photosynthetic products stored if temperature is high in winter, these photosynthetic products will be used for tree growth in summer of next year (storage effect). The positive correlation with spring temperature could due to earlier and larger snow melting which supplies the spring soil water, and eventually stimulated tree growth? Anyway, this section should be improved accordingly.
**Response: Fully accepted**. we have revised this section according you comment:
On the contrary, a significant positive response of radial growth to non-growing season temperature was found. It is possible that higher winter photosynthetic rates and more photosynthetic products stored in warm winter could be used for tree growth in summer of next year (storage effect), and higher winter temperature could also arise from less frost damage; the positive correlation with spring temperature could due to earlier and larger snow melting which supplies the spring soil water, and eventually stimulated tree growth (Hollesen et al. 2015; Zhu et al. 2017).

Line 225-226 is the "local historical record" and "historical documents" have been specified in the Data and Methods part? Since I can't find them before section 4.2.
**Response: Fully accepted**. We added the description of those data in the Methods section: Local historical drought data recorded in book "Meteorological disasters

dictionary of China" (Shen 2008; Sun 2007) were used to verify our PDSI reconstruction.

Line 234 Are PDSI reconstructions in Mengkeshan and Pangu from your data, or previous studies of other people? If they are from previous studies out- side your sampling region, is it possible to do the same SEA analysis with your own data of this study?

**Response:** In fact, the data in Mengkeshan and Pangu which were used for SEA analysis are the forest fire event data. The fire event data were reconstructed by fire scars (Yao *et al*., 2017). Forest fires usually occur in dry years, so the occurrence of forest fires can reflect drought events from the side. The SEA analysis between forest fire history and reconstructed drought variables could further validate the accuracy of our reconstruction.

Line 243-253 I agree with you that Cook's MADA reconstruction is inaccurate and sometime useless in regions with no or a few tree-ring data, such as your study region.

**Response:** Although, the divergence of between the MADA and individual tree-ring-based drought reconstruction has been found in some studies (Li *et al.*, 2015; Liu *et al.*, 2016; and this study), more evidence still need to be found.